# UniTSFace: Unified Threshold Integrated Sample-to-Sample Loss for Face Recognition

**Qiufu Li** [1,2,6,#]
liqiufu@szu.edu.cn

**Xi Jia** [1,2,3,#]
x.jia.1@cs.bham.ac.uk

**Jiancan Zhou** [1,2,4,#]
zhoujiancan@foxmail.com

**Linlin Shen** [1,2,6,*]
llshen@szu.edu.cn

**Jinming Duan** [3,5]
j.duan@bham.ac.uk

[1] National Engineering Laboratory for Big Data System Computing Technology,
Shenzhen University, China
[2] Computer Vision Institute, Shenzhen University, China
[3] School of Computer Science, University of Birmingham, UK
[4] Aqara, Lumi United Technology Co., Ltd, China
[5] Alan Turing Institute, UK
[6] SZU Branch, Shenzhen Institute of Artificial Intelligence and Robotics for Society, China

## Abstract

Sample-to-class-based face recognition models can not fully explore the cross-sample relationship among large amounts of facial images, while sample-to-sample-based models require sophisticated pairing processes for training. Furthermore, neither method satisfies the requirements of real-world face verification applications, which expect a unified threshold separating positive from negative facial pairs. In this paper, we propose a unified threshold integrated sample-to-sample based loss (USS loss), which features an explicit unified threshold for distinguishing positive from negative pairs. Inspired by our USS loss, we also derive the sample-to-sample based softmax and BCE losses, and discuss their relationship. Extensive evaluation on multiple benchmark datasets, including MFR, IJB-C, LFW, CFP-FP, AgeDB, and MegaFace, demonstrates that the proposed USS loss is highly efficient and can work seamlessly with sample-to-class-based losses. The embedded loss (USS and sample-to-class Softmax loss) overcomes the pitfalls of previous approaches and the trained facial model UniTSFace exhibits exceptional performance, outperforming state-of-the-art methods, such as CosFace, ArcFace, VPL, AnchorFace, and UNPG. Our code is available at `https://github.com/CVI-SZU/UniTSFace`.

## 1    Introduction

Modern deep facial recognition systems, involving an enormous number of facial images and identities, essentially rely on discriminative feature learning: the facial images from the same identity should be close while those from different identities should be distant in the feature space. That is, the similarity of a positive pair (two facial images from the same identity) is required to be larger than any negative pair (two facial images from different identities). In other words, a unified threshold is expected to distinguish positive from negative pairs.

The feature learning process of general deep face recognition models is either based on sample-to-class losses (such as Softmax loss (27; 28)) or sample-to-sample losses (such as contrastive (4; 10; 25)

---

[#]Equal contribution; [*]Corresponding author. Parts of the work were done when X. J and J. Z were students at Shenzhen University.

and triplet loss (22)). Inspired by the success of large-scale image classification, the softmax loss and its extensions have become popular in deep face recognition systems. In the softmax loss, each weight vector can be regarded as a proxy of the corresponding class (face identity), the classification-based face recognition models are demanded to learn the class proxy and image features simultaneously. However, these models possess a significant drawback, namely a domain gap between the training and testing stages of face models. This gap arises from the reliance on limited identity proxies for computing and optimizing feature similarity in sample-to-class models. Conversely, in real-world scenarios, feature similarity is computed and compared across various samples from diverse facial identities. In other words, the sample-to-class-based classification strategy may not entirely explore the variances across samples (**Problem 1**). Therefore, its efficacy in accurately reflecting real-world scenarios is questionable. To tackle this drawback, VPL (8) considers a small variation around the class proxy, which implicitly introduces more samples during the optimization and improves the face recognition performance. However, while this small variation extends the capability of softmax loss, it cannot essentially represent all the samples in this class.

Face models based on sample-to-sample losses (4; 10; 22; 3; 24) learn facial identity features and optimize feature similarities by comparing positive and negative samples, which is closer to real-world face recognition applications than sample-to-class losses. The majority of sample-to-sample losses aim to maximize inter-class discrepancy while minimizing intra-class distances, which may need a meticulous sampling/paring step for every mini-batch. Moreover, in face verification tasks, a single threshold is required to distinguish positive facial pairs from negative ones. Unfortunately, none of the aforementioned methods incorporate such an explicit constraint (**Problem 2**).

In this study, we commence our research by analyzing a reasonable, albeit naive, sample-to-sample loss, which enables us to straightforwardly investigate the variances across facial samples, thereby resolving **Problem 1**. To address **Problem 2**, we integrate a unified learnable threshold into the naive loss, resulting in a novel sample-to-sample loss function that we term the unified threshold integrated sample-to-sample (USS) loss. We provide a mathematical elaboration of the relationship between our USS loss and the sample-to-sample BCE loss and softmax loss, from the perspective of the naive sample-to-sample loss. To evaluate the effectiveness of the USS loss, we conduct experiments on various benchmark datasets and qualitatively demonstrate that it meets the requirements of real-world applications. Additionally, we find that incorporating a margin can easily enhance the USS loss. Our USS loss also works seamlessly with the sample-to-class based losses and shows significant improvements when they are combined together to train a face model called UniTSFace. To summarize, the contributions of this work are

- We introduce a unified threshold integrated sample-to-sample loss (USS) derived from a reasonable naive loss for face recognition. We also derive the sample-to-sample based softmax and binary-cross-entropy (BCE) losses from the naive loss and reveal the mathematical relationship among the softmax, BCE, and our USS losses.

- We demonstrate that a unified threshold can be learned by adopting the proposed USS loss and quantitatively and qualitatively demonstrate that the learned threshold aligns with face verification expectations in experiments.

- The proposed USS loss can be enhanced with an auxiliary margin and is compatible with existing sample-to-class based losses. We show that USS loss, when used jointly with sample-to-class based losses such as CosFace and ArcFace, leads to a continuous improvement.

- Our UniTSFace outperforms state-of-the-art methods on the Megaface 1 dataset and ranks first place on the MFR ongoing challenge till the submission of this work (May 17 '23, academic track): `http://iccv21-mfr.com/#/leaderboard/academic`.

## 2    Related Works

**Sample-to-Sample based Methods.**    DeepID2 (25) uses a contrastive loss(4; 10) to encourage the features learned from the same identity to be close while that learned from different identities are distant. FaceNet(22) constructs three-element tuples and minimizes the distance between an anchor and a positive sample and maximizes the distance between the anchor and a negative sample. The effectiveness of contrastive/triplet losses relies on the meticulous selection of pairs/triplets. Furthermore, neither method explicitly enforces that all positive sample-to-sample similarities are greater than negative similarities.

**Sample-to-Class based Methods.** While Softmax loss is frequently utilized in deep recognition models, it only promotes separability and does not learn discriminative features. Various approaches have been proposed to enhance the feature learning of softmax loss. Some methods (32; 9; 34; 16) proposed extra constraints imposed on the softmax loss, such methods normally have a class proxy (class center/prototype) and maximize (minimize) the similarity (distances) between a facial feature and its corresponding class proxy (32), as well as maximize the distances between all class proxies(16; 9; 34). However, the training process of such methods needs careful balancing of the softmax loss and the extra constraints. Alternatively, some works directly improve the softmax by normalizing the facial features and adding margins between positive and negative sample-to-class pairs (18; 29; 30; 7; 17). While sample-to-class methods demonstrate excellent performance in deep face verification, they may not entirely explore the variability across various facial samples.

**Hybrid (/combined) Methods.** Circle loss (26) is one of the first works that discussed the two elemental learning paradigms, i.e., learning with class-level labels and pair-wise labels, in a unified framework. However, the two learning paradigms are respectively used in circle loss. Following (26), the combination of the two learning paradigms to solve the shortcoming of either method become popular, e.g., VPL(8), AnchorFace(15), and UNPG(13). VPL extends the softmax loss by considering a small feature variation around the class proxy $\boldsymbol{W}$. The embedding of a small feature variation cannot essentially represent all samples, VPL still presents a large gap between training a face recognition model and testing over the open sets. UNPG jointly optimizes the distance between a sample $\boldsymbol{x}_i$ with negative class proxies $\boldsymbol{W}_j$ and negative samples $\boldsymbol{x}_j$ in the same softmax loss using batch processing, focusing on the negative pairs during training. AnchorFace, however, uses a combination of softmax loss and different sample-to-sample losses, such as TAR Loss and FAR Loss, to target specific testing protocols. In contrast, our USS loss is more general and compatible with existing sample-to-class losses. We demonstrate our USS loss can lead to a steady improvement when used with sample-to-class losses jointly in experiments.

## 3 Methods

Suppose $\mathcal{M}$ is a deep face model trained on a facial sample set $\mathcal{D} = \bigcup_{i=1}^{N} \mathcal{D}_i$ captured from $N$ subjects, where $\mathcal{D}_i$ denotes the subset containing the facial samples captured from the same subject $i$. Then, we can get a feature set $\mathcal{F} = \bigcup_{i=1}^{N} \mathcal{F}_i = \bigcup_{i=1}^{N} \left\{ \boldsymbol{x}^{(i)} = \mathcal{M}(\boldsymbol{X}^{(i)}) : \boldsymbol{X}^{(i)} \in \mathcal{D}_i \right\}$, where $\boldsymbol{x}^{(i)}$ is the feature vector of the sample $\boldsymbol{X}^{(i)}$ and $\boldsymbol{X}^{(i)}$ denotes the sample captured from subject $i$.

For any two samples $\boldsymbol{X}, \boldsymbol{X}_* \in \mathcal{D}$, we apply a bivariate operator $g(\boldsymbol{x}, \boldsymbol{x}_*) \in [-1, 1]$ denoting their feature similarity, where $\boldsymbol{x} = \mathcal{M}(\boldsymbol{X}), \boldsymbol{x}_* = \mathcal{M}(\boldsymbol{X}_*)$ are features of $\boldsymbol{X}, \boldsymbol{X}_*$. We term $g(\boldsymbol{x}, \boldsymbol{x}_*)$ the **positive sample-to-sample similarity** if the two samples are captured from the same subject, while the **negative sample-to-sample similarity** if they are from different subjects. Then, for any sample $\boldsymbol{X}^{(i)} \in \mathcal{D}_i, \forall i$, we define its positive and negative similarity sets as

$$\boldsymbol{s}_{\boldsymbol{X}^{(i)}}^{(\text{pos})} = \left\{ g(\boldsymbol{x}^{(i)}, \boldsymbol{x}_*) : \boldsymbol{x}_* \in \mathcal{F}_i \right\}, \tag{1}$$

$$\boldsymbol{s}_{\boldsymbol{X}^{(i)}}^{(\text{neg})} = \bigcup_{\substack{j=1 \\ j \neq i}}^{N} \left\{ g(\boldsymbol{x}^{(i)}, \boldsymbol{x}_*) : \boldsymbol{x}_* \in \mathcal{F}_j \right\}, \tag{2}$$

where $\boldsymbol{x}^{(i)} = \mathcal{M}(\boldsymbol{X}^{(i)})$.

In real applications of face verification, a threshold $\hat{t} \in [-1, 1]$ is chosen to verify whether two samples $\boldsymbol{X}$ and $\boldsymbol{X}_*$ are from the same subject or not. Specifically, the two facial samples $\boldsymbol{X}, \boldsymbol{X}_*$ are with the same identity if $g(\boldsymbol{x}, \boldsymbol{x}_*) \geq \hat{t}$, while they are from two different identities when $g(\boldsymbol{x}, \boldsymbol{x}_*) < \hat{t}$.

To be in line with face verification applications, during the training process of $\mathcal{M}$, we expect a unified threshold $t$ such that any two samples ($\boldsymbol{X}^{(i)} \in \mathcal{D}_i$ and $\boldsymbol{X}_*^{(j)} \in \mathcal{D}_j$) conform to the above rules. If they share the same identity (i.e., $i = j$), then $g(\boldsymbol{x}^{(i)}, \boldsymbol{x}_*^{(j)}) \geq t$, in the case of $i \neq j$, $g(\boldsymbol{x}^{(i)}, \boldsymbol{x}_*^{(j)}) < t$, where $\boldsymbol{x}^{(i)} = \mathcal{M}(\boldsymbol{X}^{(i)}), \boldsymbol{x}_*^{(j)} = \mathcal{M}(\boldsymbol{X}_*^{(j)})$. The unified threshold $t$ satisfies

$$\max \left( \bigcup_{i=1}^{N} \bigcup_{\boldsymbol{X}^{(i)} \in \mathcal{D}_i} \boldsymbol{s}_{\boldsymbol{X}^{(i)}}^{(\text{neg})} \right) < t \leq \min \left( \bigcup_{i=1}^{N} \bigcup_{\boldsymbol{X}^{(i)} \in \mathcal{D}_i} \boldsymbol{s}_{\boldsymbol{X}^{(i)}}^{(\text{pos})} \right). \tag{3}$$

However, the existing sample-to-sample losses (4; 10; 22; 24) fail to explicitly include and learn the unified threshold. In this paper, by explicitly defining the unified threshold $t$, we design a unified threshold integrated sample-to-sample loss.

### 3.1 Unified Threshold Integraed Sample-to-Sample Loss (USS Loss)

Clearly, in the training of model $\mathcal{M}$, it expects large positive sample-to-sample similarities but small negative ones, and then, for any sample $\boldsymbol{X}^{(i)}$, a naive loss could be reasonably defined as,

$$L_{\text{NAIVE}}(\boldsymbol{X}^{(i)}) = -\frac{1}{|\mathcal{F}_i|} \sum_{\boldsymbol{x} \in \mathcal{F}_i} \gamma g(\boldsymbol{x}^{(i)}, \boldsymbol{x}) + \frac{1}{|\mathcal{F} - \mathcal{F}_i|} \sum_{\substack{j=1 \\ j \neq i}}^{N} \sum_{\boldsymbol{x} \in \mathcal{F}_j} \gamma g(\boldsymbol{x}^{(i)}, \boldsymbol{x}), \quad (4)$$

where $\gamma$ is a scale factor, $|\mathcal{S}|$ denotes the element number of a set $\mathcal{S}$, and $\mathcal{F} - \mathcal{F}_i$ denotes the complementary set of $\mathcal{F}_i$ in $\mathcal{F}$, i.e., $\mathcal{F} - \mathcal{F}_i = \bigcup_{\substack{j=1 \\ j \neq i}}^{N} \mathcal{F}_j$.

The loss in Eq. (4) computes the feature similarities of all positive sample-to-sample pairs and all negative ones for the sample $\boldsymbol{X}^{(i)}$, which is not easily implemented in practice. In this paper, without loss of generality, we consider the feature similarities of one positive sample pair and $N - 1$ negative sample pairs for the sample $\boldsymbol{X}^{(i)}$ in the naive loss,

$$L_{\text{naive}}(\boldsymbol{X}^{(i)}) = -\gamma g(\boldsymbol{x}^{(i)}, \boldsymbol{x}_*^{(i)}) + \frac{1}{N-1} \sum_{\substack{j=1 \\ j \neq i}}^{N} \gamma g(\boldsymbol{x}^{(i)}, \boldsymbol{x}_*^{(j)}), \quad (5)$$

where $\boldsymbol{x}_*^{(i)} = \mathcal{M}(\boldsymbol{X}_*^{(i)})$, $\boldsymbol{x}_*^{(j)} = \mathcal{M}(\boldsymbol{X}_*^{(j)})$, and $\boldsymbol{X}_*^{(i)}$ and $\boldsymbol{X}_*^{(j)}$ are randomly taken from the subject $i, j$, with $j \neq i$.

Using the inequality of arithmetic and geometric means[1], we derive two inequalities about $L_{\text{naive}}$,

$$L_{\text{naive}}(\boldsymbol{X}^{(i)}) \leq 2 \log \left( 1 + \frac{\exp \left( \sum_{\substack{j=1 \\ j \neq i}}^{N} \frac{\gamma g(\boldsymbol{x}^{(i)}, \boldsymbol{x}_*^{(j)})}{N-1} \right)}{\exp \left( \gamma g(\boldsymbol{x}^{(i)}, \boldsymbol{x}_*^{(i)}) \right)} \right) - 2 \log 2, \quad (6)$$

$$L_{\text{naive}}(\boldsymbol{X}^{(i)}) \leq \frac{2}{N-1} \sum_{\substack{j=1 \\ j \neq i}}^{N} \log \left( 1 + \frac{e^{\gamma g(\boldsymbol{x}^{(i)}, \boldsymbol{x}_*^{(j)})}}{e^{\gamma g(\boldsymbol{x}^{(i)}, \boldsymbol{x}_*^{(i)})}} \right) - 2 \log 2. \quad (7)$$

According to Eqs. (6), (7), and the unified threshold $t$ in Eq. (3), one can get

$$\frac{N}{2} L_{\text{naive}}(\boldsymbol{X}^{(i)}) + N \log 2$$

$$\leq \log \left( 1 + \frac{\exp \left( \sum_{\substack{j=1 \\ j \neq i}}^{N} \frac{\gamma g(\boldsymbol{x}^{(i)}, \boldsymbol{x}_*^{(j)})}{N-1} \right)}{\exp \left( \gamma g(\boldsymbol{x}^{(i)}, \boldsymbol{x}_*^{(i)}) \right)} \right) + \sum_{\substack{j=1 \\ j \neq i}}^{N} \log \left( 1 + \frac{\exp(\gamma g(\boldsymbol{x}^{(i)}, \boldsymbol{x}_*^{(j)}))}{\exp(\gamma g(\boldsymbol{x}^{(i)}, \boldsymbol{x}_*^{(i)}))} \right) \quad (8)$$

$$\leq \log \left( 1 + \frac{\exp \left( \sum_{\substack{j=1 \\ j \neq i}}^{N} \frac{\gamma t}{N-1} \right)}{\exp \left( \gamma g(\boldsymbol{x}^{(i)}, \boldsymbol{x}_*^{(i)}) \right)} \right) + \sum_{\substack{j=1 \\ j \neq i}}^{N} \log \left( 1 + \frac{\exp(\gamma g(\boldsymbol{x}^{(i)}, \boldsymbol{x}_*^{(j)}))}{\exp(\gamma t)} \right) \quad (9)$$

$$= \log \left( 1 + e^{-\gamma g(\boldsymbol{x}^{(i)}, \boldsymbol{x}_*^{(i)}) + \gamma t} \right) + \sum_{\substack{j=1 \\ j \neq i}}^{N} \log \left( 1 + e^{\gamma g(\boldsymbol{x}^{(i)}, \boldsymbol{x}_*^{(j)}) - \gamma t} \right). \quad (10)$$

We define the **U**nified threshold integrated **S**ample-to-**S**ample (USS) loss $L_{\text{uss}}(\boldsymbol{X}^{(i)})$ as

$$L_{\text{uss}}(\boldsymbol{X}^{(i)}) = \log \left( 1 + e^{-\gamma g(\boldsymbol{x}^{(i)}, \boldsymbol{x}_*^{(i)}) + b} \right) + \sum_{\substack{j=1 \\ j \neq i}}^{N} \log \left( 1 + e^{\gamma g(\boldsymbol{x}^{(i)}, \boldsymbol{x}_*^{(j)}) - b} \right) \quad (11)$$

---

[1] $\sqrt[n]{\prod_{i=1}^{n} a_i} \leq \frac{1}{n} \sum_{i=1}^{n} a_i$ for $a_i \geq 0$.

where $b = \gamma t$ is a constant to be learned (depicted in Fig. 1). The detailed derivations of the above inequalities are described in the supplementary (appendix).

We here analyze that the unified threshold $t$ could be learned. Suppose that the model $\mathcal{M}$ has been perfectly trained, and $L_{\text{uss}}$ has reached its minimum point after the training, which means (i) the positive sample-to-sample similarity $g(\boldsymbol{x}^{(i)}, \boldsymbol{x}_*^{(i)})$ tends to 1, and the negative ones $g(\boldsymbol{x}^{(i)}, \boldsymbol{x}_*^{(j)})$ tends to $-1$; and (ii) $L_{\text{uss}}$ reaches its stationary point in terms of variable $b$. From (ii), one can deduce that

$$0 = \frac{\partial L_{\text{uss}}}{\partial b} = \frac{\mathrm{e}^{-\gamma g(\boldsymbol{x}^{(i)}, \boldsymbol{x}_*^{(i)}) + b}}{1 + \mathrm{e}^{-\gamma g(\boldsymbol{x}^{(i)}, \boldsymbol{x}_*^{(i)}) + b}} - \sum_{\substack{j=1 \\ j \neq i}}^{N} \frac{\mathrm{e}^{\gamma g(\boldsymbol{x}^{(i)}, \boldsymbol{x}_*^{(j)}) - b}}{1 + \mathrm{e}^{\gamma g(\boldsymbol{x}^{(i)}, \boldsymbol{x}_*^{(j)}) - b}} \tag{12}$$

$$\overset{\text{(i)}}{=} \frac{\mathrm{e}^{-\gamma + b}}{1 + \mathrm{e}^{-\gamma + b}} - \sum_{\substack{j=1 \\ j \neq i}}^{N} \frac{\mathrm{e}^{-\gamma - b}}{1 + \mathrm{e}^{-\gamma - b}} \tag{13}$$

$$\Rightarrow b = \log \frac{(N-2)\mathrm{e}^{-\gamma} + \sqrt{(N-2)^2 \mathrm{e}^{-2\gamma} + 4(N-1)}}{2}. \tag{14}$$

If $N < \frac{\mathrm{e}^{2\gamma} + 3}{2}$, the final learned threshold $t = \frac{b}{\gamma}$ will locate between $-1$ and 1, which means now the leaned $t$ has correctly separated the negative sample-to-sample similarities, $g(\boldsymbol{x}^{(i)}, \boldsymbol{x}_*^{(j)})$, and the positive ones, $g(\boldsymbol{x}^{(i)}, \boldsymbol{x}_*^{(i)})$. In the practice, following (30; 7), we set $\gamma = 64$, then, according to the above analysis, the unified threshold $t$ could be leaned if $N < 1.9 \times 10^{55}$.

## 3.2 Sample-to-Sample Based Softmax and BCE Losses

Sample-to-class based softmax and BCE losses are widely applied in the image classification. For the study of face verification, we here deduce the sample-to-sample based softmax and BCE losses from the naive loss $L_{\text{naive}}$.

Similar to the deduction of USS loss, using the inequality of arithmetic and geometric means, we first present another two inequalities about $L_{\text{naive}}$,

$$L_{\text{naive}}(\boldsymbol{X}^{(i)}) \leq -\frac{N}{N-1} \log \frac{\mathrm{e}^{\gamma g(\boldsymbol{x}^{(i)}, \boldsymbol{x}_*^{(i)})}}{\sum_{j=1}^{N} \mathrm{e}^{\gamma g(\boldsymbol{x}^{(i)}, \boldsymbol{x}_*^{(j)})}} - \frac{N \log N}{N-1}, \tag{15}$$

$$\sum_{i=1}^{N} L_{\text{naive}}(\boldsymbol{X}^{(i)}) \leq \frac{2}{N-1} \sum_{i=1}^{N} \sum_{\substack{j=1 \\ j \neq i}}^{N} \log \left(1 + \frac{\mathrm{e}^{\gamma g(\boldsymbol{x}^{(i)}, \boldsymbol{x}_*^{(j)})}}{\mathrm{e}^{\gamma g(\boldsymbol{x}^{(j)}, \boldsymbol{x}_*^{(j)})}}\right) - 2N \log 2. \tag{16}$$

**Softmax Loss.** For sample $\boldsymbol{X}^{(i)} \in \mathcal{D}_i$ with $\boldsymbol{x}^{(i)} = \mathcal{M}(\boldsymbol{X}^{(i)})$, we define its sample-to-sample softmax loss as

$$L_{\text{soft}}(\boldsymbol{X}^{(i)}) = -\log \frac{\mathrm{e}^{\gamma g(\boldsymbol{x}^{(i)}, \boldsymbol{x}_*^{(i)})}}{\sum_{j=1}^{N} \mathrm{e}^{\gamma g(\boldsymbol{x}^{(i)}, \boldsymbol{x}_*^{(j)})}}. \tag{17}$$

Then, according to Eq. (15), one can get

$$L_{\text{naive}}(\boldsymbol{X}^{(i)}) \leq \frac{N}{N-1} L_{\text{soft}}(\boldsymbol{X}^{(i)}) - \frac{N}{N-1} \log N. \tag{18}$$

Similar to the naive loss $L_{\text{naive}}$, the design of softmax loss $L_{\text{soft}}$ does not consider the unified threshold among the sample-to-sample pairs.

**BCE Loss.** For all samples captured from subject $i$, we assume the existence of a threshold $t_i$, which could separate their all positive sample-to-sample pairs and negative ones, i.e.,

$$\max\left(\bigcup_{\boldsymbol{X}^{(i)} \in \mathcal{D}_i} \boldsymbol{s}_{\boldsymbol{X}^{(i)}}^{\text{(neg)}}\right) < t_i \leq \min\left(\bigcup_{\boldsymbol{X}^{(i)} \in \mathcal{D}_i} \boldsymbol{s}_{\boldsymbol{X}^{(i)}}^{\text{(pos)}}\right), \quad \forall i, \tag{19}$$

then, according to Eqs. (16) and (6),

$$\frac{N}{2}\sum_{i=1}^{N} L_{\text{naive}}(\boldsymbol{X}^{(i)}) + N^2 \log 2$$

$$\leq \sum_{i=1}^{N}\left[\log\left(1 + \frac{\exp\left(\sum_{\substack{j=1\\j\neq i}}^{N}\frac{\gamma g(\boldsymbol{x}^{(i)},\boldsymbol{x}_*^{(j)})}{N-1}\right)}{\exp\left(\gamma g(\boldsymbol{x}^{(i)},\boldsymbol{x}_*^{(i)})\right)}\right) + \sum_{\substack{j=1\\j\neq i}}^{N}\log\left(1 + \frac{\exp(\gamma g(\boldsymbol{x}^{(i)},\boldsymbol{x}_*^{(j)}))}{\exp(\gamma g(\boldsymbol{x}^{(j)},\boldsymbol{x}_*^{(j)}))}\right)\right] \qquad (20)$$

$$\leq \sum_{i=1}^{N}\left[\log\left(1 + e^{-\gamma g(\boldsymbol{x}^{(i)},\boldsymbol{x}_*^{(i)})+\gamma t_i}\right) + \sum_{\substack{j=1\\j\neq i}}^{N}\log\left(1 + e^{\gamma g(\boldsymbol{x}^{(i)},\boldsymbol{x}_*^{(j)})-\gamma t_j}\right)\right]. \qquad (21)$$

We define BCE loss for sample $\boldsymbol{X}^{(i)}$ as

$$L_{\text{bce}}(\boldsymbol{X}^{(i)}) = \log\left(1 + e^{-\gamma g(\boldsymbol{x}^{(i)},\boldsymbol{x}_*^{(i)})+b_i}\right) + \sum_{\substack{j=1\\j\neq i}}^{N}\log\left(1 + e^{\gamma g(\boldsymbol{x}^{(i)},\boldsymbol{x}_*^{(j)})-b_j}\right), \qquad (22)$$

where $b_i = \gamma t_i$ are parameters to be learned. Note that the $t_i$ can be different for different identities and therefore are not unified.

After the training of the model $\mathcal{M}$, i.e., the threshold $t_i = \frac{b_i}{\gamma}$ were learned, then,

$$\sum_{i=1}^{N} L_{\text{naive}}(\boldsymbol{X}^{(i)}) \leq \frac{2}{N}\sum_{i=1}^{N} L_{\text{bce}}(\boldsymbol{X}^{(i)}) - 2N\log 2. \qquad (23)$$

### 3.3 Marginal Sample-to-Sample Based Losses

Our USS loss, as well as the deduced sample-to-sample based $L_{\text{soft}}$ and $L_{\text{bce}}$, only encourage the separability between positive and negative sample pairs. To further improve the discriminative ability of such losses, i.e., to encourage the positive features to be distant from the negative ones, we further proposed the marginal extensions for such losses.

By introducing a margin on the feature similarities, we can have the marginal USS loss as:

$$L_{\text{uss-m}}(\boldsymbol{X}^{(i)}) = \log\left(1 + e^{-\gamma(g(\boldsymbol{x}^{(i)},\boldsymbol{x}_*^{(i)})-m)+b}\right) + \sum_{\substack{j=1\\j\neq i}}^{N}\log\left(1 + e^{\gamma g(\boldsymbol{x}^{(i)},\boldsymbol{x}_*^{(j)})-b}\right) \qquad (24)$$

where the $m$ is the introduced hyper-parameter on the margin. Proper adjusting of such a parameter can improve the recognition performance, as discussed in Sec. 4.3. Similarly, we can extend the vanilla $L_{\text{soft}}$ and $L_{\text{bce}}$ to the marginal version of $L_{\text{soft-m}}$ and $L_{\text{bce-m}}$, the full equations are given in the supplementary materials (and appendix).

## 4 Experiments

### 4.1 Datasets and Evaluations

**Datasets.** We utilize four publicly available datasets for training, namely, CASIA-WebFace(33) (consisting of 0.5 million images of 10K identities), Glint360K(2) (comprising 17.1 million images of 360K identities), WebFace42M(35) (containing 42.5 million images of 2 million identities), and WebFace4M, which is a subset of WebFace42M with 4.2 million images of 0.2 million identities. For evaluating the face verification performance, the ICCV-2021 Masked Face Recognition Challenge (MFR Ongoing)(5) is adopted. The MFR ongoing testing protocol includes various popular benchmarks, such as LFW (11), CFP-FP (23), AgeDB (21), and IJB-C (20), along with its own MFR benchmarks, such as the Mask, Children, and Globalized Multi-Racial (GMR) test sets. The Mask set comprises 13.9K positive pairs and 96.9 million negative pairs (including 6.9K masked images and

13.9K non-masked images) of 6.9K identities. The Children set contains 157K images (totaling 1.7 million positive pairs and 24.7 billion negative pairs) of 14K identities. The Globalized Multi-Racial sets comprise 1.6 million images in total, consisting of 4.6 million positive pairs and 2.6 trillion negative pairs, and representing 242K identities across four races: African, Caucasian, South-Asian, and East-Asian. For face identification, the MegaFace Challenge 1(14) is employed as the test set, which comprises a gallery set with over 1 million images from 690K different identities, and a probe set with 3,530 images from 530 identities. It is worth noting that the MegaFace Challenge 1 also has a verification track, and the verification performance is included in the experiments.

**Evaluation and Metrics.** For the MFR Ongoing Challenge, the trained models are submitted to and evaluated by the online server. Specifically, we report 1:1 verification accuracy for LFW, CFP-FP, and AgeDB. We report True Accept Rate (TAR) at False Accept Rate (FAR) levels of 1e-4 and 1e-5 for IJB-C. We report TARs at FAR=1e-4 for the Mask and Children test sets, and TARs at FAR=1e-6 for the GMR test sets. For the MegaFace Challenge 1, we report Rank1 accuracy for identification and TAR at FAR=1e-6 for verification.

## 4.2 Implementation Details

**Preprocessing**. Firstly, we aligned all face images using the 5 landmarks detected by RetinaFace (6) and cropped the center $112 \times 112$ patch. We then normalized the cropped images by first subtracting 127.5 and then dividing 128. Finally, we augmented the training images with horizontal flipping.

**Training**. We adopt customized ResNets as our backbone following (7). We implement all models using Pytorch and train them using the SGD optimizer with a weight decay of 5e-4 and momentum of 0.9. For the face models on CASIA-WebFace, we train them over 28 epochs with a batch size of 512. The learning rate starts at 0.1 and is reduced by a factor of 10 at the $16^{th}$ and $24^{th}$ epoch. For both Glint360K and WebFace4M, we train the ResNets for 20 epochs using a batch size of 1024. The learning rate is initially set at 0.1, while a polynomial decay strategy (power=2) is applied for the learning rate schedule. In the case of WebFace42M, we train the ResNets for 20 epochs, using a larger batch size of 4096. The learning rate linearly warms up from 0 to 0.4 during the first epoch, followed by a polynomial decay (power=2) for the remaining 19 epochs. We include the detailed settings of all hyper-parameters used in Sec. 4 and Sec. 5 in the appendix for further reference.

**Testing**. For a given facial image, we extract two 512-dimensional features from the original image and its horizontally flipped counterpart. These features are then combined together as the final representation with element-wise addition. To assess the similarity between two images, we utilize the cosine similarity metric.

## 4.3 Ablation and Parameter Study

**Effectiveness of unified threshold.** We first demonstrate the necessity of learning a unified threshold by comparing our proposed $L_{uss}$ with two other sample-to-sample losses, $L_{soft}$ and $L_{bce}$, in Table 1. We additionally report the performance of the model trained with $L_{naive}$. The first to third rows respectively show the results from $L_{naive}$, $L_{soft}$ and $L_{bce}$, while the fourth row lists the performance of our $L_{uss}$. We observe that $L_{soft}$ performs $4.68\%$ better than our proposed $L_{uss}$ on IJB-C. However, our USS loss outperforms $L_{soft}$ on the other four

| Loss | MR-ALL | IJB-C | LFW | CFP | Age |
|------|--------|-------|-----|-----|-----|
| $L_{naive}$ | 0.0 | 0.35 | 50.0 | 50.0 | 50.0 |
| $L_{soft}$ | 26.34 | **76.88** | 98.80 | 95.50 | 93.48 |
| $L_{bce}$ | 8.91 | 39.19 | 92.60 | 66.41 | 74.36 |
| $L_{uss}$ | **38.43** | 72.20 | **99.40** | **96.51** | **94.05** |
| $L_{soft-m}$ | 38.11 | 83.60 | 99.18 | 96.32 | 94.18 |
| $L_{bce-m}$ | 9.01 | 45.08 | 92.48 | 65.34 | 76.30 |
| $L_{uss-m}$ | **42.55** | **83.92** | **99.46** | **96.81** | **94.28** |

Table 1: Ablation study of the proposed $L_{uss}$.

datasets, particularly on MFR-All where it achieves a $12.09\%$ higher TAR@FAR=1e-6. Compared to the $L_{bce}$, the gains achieved by our $L_{uss}$ are more significant, with performance improvements of $29.52\%$ on MFR-All, $33.01\%$ in terms of TAR@FAR=1e-4 on IJB-C, $30.10\%$ on CFP-FP, and $19.69\%$ on AgeDB. The $L_{naive}$, however, is difficult to train and the trained model barely learns any useful features. We note that the results are not cherry-picked. For a fair comparison, we adopted the same experimental setting when we trained the model using the different losses. During our experiments, we found that the model is easy to converge using $L_{soft}$ and $L_{uss}$, and the training process is stable, which leads to more favorable results. The model, however, is difficult to train with $L_{naive}$ and $L_{bce}$. We have dedicated many efforts to these two losses, but the convergence is still problematic.

When comparing the marginal softmax loss $L_{\text{soft-m}}$, marginal BCE loss $L_{\text{bce-m}}$ with marginal USS loss $L_{\text{uss-m}}$ (last three rows), we observe that the marginal USS loss $L_{\text{uss-m}}$ achieves the highest performance across all five reported datasets, which further confirms the effectiveness of our proposed unified threshold strategy.

It is noteworthy that all the methods in this table are based on the same ResNet-50 network trained on the CASIA-WebFace dataset.

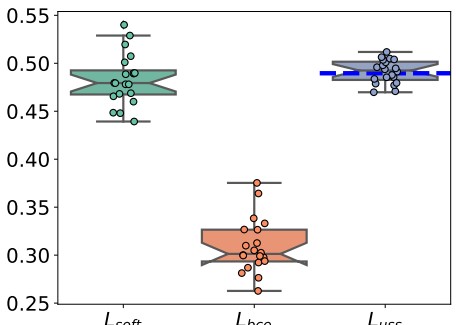

Figure 1: Threshold distributions of $L_{\text{soft}}$, $L_{\text{bce}}$, and $L_{\text{uss}}$ on 20 random selected identities in CASIA-WebFace. Each dot denotes the optimal threshold for each identity. The dashed blue line represents the threshold that was learned solely by $L_{\text{uss}}$ with $t = \frac{b}{\gamma} = 31.3344/64 = 0.4896$ in Eq. (11).

**Qualitative study of unified threshold.** To better understand the effects of unified threshold, in Fig. 1, we randomly select 20 identities (5,074 facial images) from the whole training dataset with $N = 10572$ subjects, and then construct 1 positive sample pair and $N - 1$ negative pairs for each of the selected images. For each of the 20 identities, we compute the optimal threshold to separate the positive pair and the hardest negative pair, i.e., the most similar negative pair. We respectively plot threshold distributions of the 20 identities for $L_{\text{soft}}$, $L_{\text{bce}}$, and $L_{\text{uss}}$. In Fig. 1, each dot denotes the optimal threshold for each identity. The blue dashed line is the unified threshold $t = \frac{b}{\gamma} = 31.3344/64 = 0.4896$ exclusively learned by our $L_{\text{uss}}$ in Eq. (11). From this figure, we can clearly observe that our $L_{\text{uss}}$ has the most compact threshold distribution, while the threshold distributions from $L_{\text{bce}}$ and $L_{\text{soft}}$ are relatively loose. Most importantly, for $L_{\text{uss}}$, the median threshold from the randomly sampled 20 identities is in line with the learned unified threshold $0.4896$ from the face model trained with $L_{\text{uss}}$, which proves the effects of including an explicit unified threshold.

**Impact of Margins.** The results in Table 1 (with $m = 0.1$) demonstrate that a margin improves the performance of $L_{\text{uss}}$. However, the introduced margin $m$ remains a hyperparameter that needs to be tuned. In Table 2, we investigate the effects of using different margins in $L_{\text{uss-m}}$. Specifically, we experiment with four more settings that increase $m$ to 0.2, 0.3, 0.4, and 0.5. We observe that i) the highest performance on MR-ALL, IJB-C, LFT, CFP-FP, and AgeDB are respectively achieved when $m$ equals 0.2, 0.4, 0.1, 0.1, and 0.3; ii) the performance of the four marginal $L_{\text{uss-m}}$ losses are all higher than that of the original $L_{\text{uss}}$ (i.e., $m = 0$ in $L_{\text{uss-m}}$). Although the auxiliary margin improves the overall performance, the performance seems to saturate when $m$ is set to 0.4. Note that in our later experiments, we use $m = 0.1$.

| Margin | MR-ALL | IJB-C | LFW | CFP | Age |
|---|---|---|---|---|---|
| $m = 0.0$ | 38.43 | 72.20 | 99.40 | 96.51 | 94.05 |
| $m = 0.1$ | 42.55 | 83.92 | **99.46** | **96.81** | 94.28 |
| $m = 0.2$ | **44.74** | 87.65 | 99.26 | 96.65 | 94.16 |
| $m = 0.3$ | 44.14 | 87.58 | 99.30 | 96.41 | **94.38** |
| $m = 0.4$ | 43.38 | **87.82** | 99.26 | 96.04 | 94.23 |
| $m = 0.5$ | 43.91 | 87.62 | 99.13 | 96.05 | 94.23 |

Table 2: Parameter study of margin in $L_{\text{uss-m}}$.

| | MR-ALL | IJB-C | LFW | CFP | Age |
|---|---|---|---|---|---|
| USS | 42.55 | 83.92 | 99.46 | 96.81 | 94.28 |
| ArcFace | 42.21 | 48.49 | 99.31 | 97.07 | 94.51 |
| ArcFace+USS | **48.92** | **89.56** | **99.40** | 97.22 | **95.20** |
| CosFace | 45.12 | 56.65 | 99.36 | 97.30 | 94.98 |
| CosFace+USS | **50.28** | **89.84** | **99.41** | 97.35 | **95.13** |

Table 3: Combination of USS and two sample-to-class based methods.

**Compatibility.** In Table 1 and Table 2, we have shown that the proposed $L_{\text{uss}}$ is superior to other sample-to-sample based losses and can be boosted by adding a proper margin. In Table 3, we investigate the effects of combining $L_{\text{uss}}$ with two marginal softmax losses, i.e., ArcFace and CosFace. We find that the model trained with the combined loss significantly outperforms the original ArcFace and CosFace models, as well as our USS loss. For example, the model trained with ArcFace + USS respectively outperforms ArcFace and USS with 6.71% and 6.37% gains on MR-ALL, and 41.07% and 5.64% gains on IJB-C. The model trained with CosFace + USS respectively outperforms CosFace and USS with 5.16% and 7.73% gains on MR-ALL, and 33.19% and 5.92% gains on IJB-C. Such significant improvements suggest the compatibility of our USS loss. In this work, we refer to the model trained using a combination of a sample-to-class method and our USS as UniTSFace. For our experiments, we choose to use CosFace as the sample-to-class method.

| Method | P | R | Iden. | Veri. | Method | P | R | Iden. | Veri. |
|---|---|---|---|---|---|---|---|---|---|
| Softmax Loss (18) | S | ✗ | 54.85 | 65.92 | ArcFace (7) | L | ✗ | 81.03 | 96.98 |
| Triplet Loss (18; 22) | S | ✗ | 64.79 | 78.32 | CurricularFace (12) | L | ✗ | 81.26 | 97.26 |
| Contrastive (18; 25) | S | ✗ | 65.21 | 78.86 | CosFace (30) | L | ✗ | 82.72 | 96.65 |
| Center (18; 32) | S | ✗ | 65.49 | 80.14 | UniTSFace | L | ✗ | **85.01** | **97.85** |
| L-Softmax (18; 19) | S | ✗ | 67.12 | 80.42 | SphereFace2 (31) | L | ✓ | 89.84 | 91.94 |
| SphereFace (18) | S | ✗ | 72.72 | 85.56 | CosFace (7; 30) | L | ✓ | 97.91 | 97.91 |
| SphereFace+ (16) | S | ✗ | 73.03 | - | SphereFace (18) | L | ✓ | 98.16 | 98.46 |
| CosFace (30) | S | ✗ | 77.11 | 89.88 | ArcFace (7) | L | ✓ | 98.35 | 98.48 |
| ArcFace (7) | S | ✗ | 77.50 | 92.34 | Circle Loss (26) | L | ✓ | 98.50 | 98.73 |
| CurricularFace (12) | S | ✗ | **77.65** | 92.91 | CurricularFace (12) | L | ✓ | 98.71 | 98.64 |
| UniTSFace | S | ✗ | 77.41 | **93.50** | VPL (8) | L | ✓ | 98.80 | 98.97 |
| ArcFace (7) | S | ✓ | 91.75 | 93.69 | Partial FC (2) | L | ✓ | 98.94 | 99.10 |
| CurricularFace (12) | S | ✓ | **92.48** | 94.55 | UNPG (13) | L | ✓ | **99.27** | - |
| UniTSFace | S | ✓ | 92.36 | **94.84** | UniTSFace | L | ✓ | **99.27** | **99.19** |

Table 4: Comparisons between different methods on the MegaFace Challenge 1. The letter **P** indicates the 'Small' or 'Large' protocols, **R** denotes whether the label refinement is used. All reported results, with the exception of our own, were directly taken from their respective papers.

| Method | Net. Data. | MFR | | | | | | | IJB-C | | Verification Acc. | | |
|---|---|---|---|---|---|---|---|---|---|---|---|---|---|
| | | Mask | Child. | Afri. | Cau. | S-A. | E-A. | MR-All | 1e-4 | 1e-5 | LFW | CFP | Age |
| Contrastive(4) | | 6.67 | 10.58 | 12.40 | 18.84 | 13.57 | 10.38 | 12.38 | 58.47 | 46.75 | 95.50 | 74.65 | 82.28 |
| (N+1)-Tuplet (24) | | 26.56 | 28.23 | 39.16 | 50.92 | 47.71 | 24.06 | 38.11 | 83.60 | 73.93 | 99.18 | 96.32 | 94.18 |
| ArcFace(7) | | 38.52 | 31.42 | 45.87 | 63.69 | **59.85** | 7.66 | 42.21 | 48.49 | 9.18 | 99.31 | 97.07 | 94.51 |
| CosFace(30) | | **38.79** | 31.33 | 48.06 | 63.56 | 58.71 | 15.08 | 45.12 | 56.65 | 11.30 | 99.36 | 97.30 | 94.98 |
| Sphere-Rv1(17) | R50 | 32.80 | 28.09 | 40.24 | 57.24 | 50.38 | 22.30 | 39.92 | 86.35 | 75.81 | 99.38 | 96.95 | 94.48 |
| SphereFace2(31) | CASIA | 35.40 | 30.55 | 46.65 | 62.69 | 56.23 | 26.65 | 44.20 | 88.41 | 79.18 | 99.46 | 97.42 | 94.96 |
| VPL(8) | | 33.86 | 31.39 | 46.52 | 59.93 | 54.07 | 27.18 | 47.02 | 88.44 | 81.38 | 99.30 | 97.07 | 94.75 |
| AnchorFace(15) | | 37.04 | 32.28 | 49.60 | 63.17 | 59.80 | 28.88 | 48.44 | 88.81 | 77.82 | **99.56** | **97.48** | **95.18** |
| UNPG(13) | | 38.62 | **33.24** | 49.94 | 63.85 | 59.60 | 29.21 | 48.66 | 88.17 | 77.73 | 99.45 | 97.25 | 94.83 |
| UniTSFace | | 37.98 | 31.73 | **51.45** | **64.89** | 59.73 | **29.56** | **50.28** | **89.84** | **82.64** | 99.41 | 97.35 | 95.13 |
| Partial FC (1) | R50 | 72.28 | - | 84.86 | 91.57 | 88.57 | 67.52 | 86.85 | - | - | - | - | - |
| UniTSFace | WF4M | 75.93 | 72.00 | 88.17 | 93.68 | 91.40 | 70.55 | 89.65 | 97.03 | 95.18 | 99.80 | 99.04 | 97.93 |
| Partial FC (1) | R200 | 91.87 | - | 97.79 | 98.70 | 98.54 | 89.52 | 97.70 | 97.97 | 96.93 | **99.83** | **99.51** | 98.70 |
| UniTSFace | WF42M | **92.87** | **93.51** | **98.35** | **99.03** | **98.99** | 90.76 | **98.16** | **97.99** | **97.00** | **99.83** | 99.47 | **98.71** |

Table 5: Comparisons between different methods on MFR-Ongoing.

# 5 Comparison with the State-Of-The-Art

**MegaFace Challenge 1.** In Table 4, we compare both the identification and verification performance of our UniTSFace with several state-of-the-art methods on MegaFace Challenge 1. These methods include sample-to-sample based Contrastive Loss (4; 10) and Triplet Loss (22), sample-to-class based CosFace(30) and ArcFace(7), as well as the hybrid/combined approaches: VPL(8) and UNPG(13).

To ensure a fair comparison, as per the official protocols, we compare our UniTSFace trained on the CASIA-WebFace with the models trained on 'Small' datasets, while UniTSFace trained on Glint360K is compared with the models trained on 'Large' datasets in Table 4. When label refinement (7) is not used, our UniTSFace achieves the highest accuracy among the compared models trained on 'Large' datasets, with an 85.01% identification accuracy and a 97.85% verification accuracy. When label refinement (7) is used, the accuracies can be further increased to 99.27% and 99.19% respectively. Though our UniTSFace is slightly lower than CurricularFace (12) in terms of identification performance on 'Small' datasets, UniTSFace achieves the highest accuracy on the verification track, regardless of whether label refinement is used or not. These results demonstrate the effectiveness of our UniTSFace in both identification and verification tasks.

**MFR Ongoing Benchmarks.** We then compare the proposed UniTSFace with Contrastive Loss(4; 10), (N+1)-Tuplet Loss(24), CosFace(30), ArcFace(7), VPL(8), AnchorFace(15), and UNPG(13) on the MFR Ongoing benchmark. We re-implement these methods with the optimal hyper-parameters recommended in their original papers. All the compared models are trained with a ResNet-50 backbone and the CASIA-WebFace dataset. As reported in Table 5, our UniTSFace achieves clear improvement over other methods.

Additionally, we compare our UniTSFace with the recent Partial FC(1), which is the leading method on the MFR ongoing challenge. Following the settings of Partial FC(1), we train the proposed UniTSFace on the WebFace4M and WebFace42M datasets using two different architectures, i.e., ResNet-50 and ResNet-200. We can observe that, on average, our performance is better than that of the Partial FC. Till the submission of this work (May 17 '23), the proposed UniTSFace ranks first place on the academic track of the MFR-ongoing leaderboard.

# 6   Discussion and Conclusion

**Discussion.**

1) Though the threshold range of USS is narrowed compared to other losses in Fig. 1, is it correct to claim the word "unified"? Firstly, our theoretical objective is to learn a unified threshold that satisfies Eq.(3) during the training of a facial model. Therefore, we first assume the existence of such a unified threshold, and then propose the unified threshold integrated sample-to-sample (USS) loss. We have proven through our analysis in Section 3.1 that, ideally, a model trained by USS could learn a unified threshold for the training dataset.

However, we must admit that achieving this ideal goal is subjective to the model capacity, training hyper-parameters, and even the training dataset itself, which are all independent from our USS loss. For example, if the backbone network only uses one single linear neural layer, our USS loss definitely cannot guarantee a unified threshold either. In Fig. 1, using the same backbone architecture and training hyper-parameters, i) our USS loss is able to achieve a more compact threshold distribution than the other losses, moreover, and ii) the learned threshold denoted by the blue dashed line lies around the median of the boxplot, these two observations suggest the superiority of imposing the unified threshold and are consistent with our expectations.

2) In the testing stage, how to determine the threshold? The threshold learned by the training stage cannot be directly used in testing. In the testing stage, the threshold is determined according to the specific testing criteria. For example, when reporting the 1:1 verification accuracy on LFW, CFP-FP, AgeDB, 10-fold validation is used. We first select the threshold that achieves the highest accuracy in the first 9 folds and then adopt this threshold to calculate the accuracy in the leave-out fold.

3) How does UniTSFace compare to other methods such as CosFace in terms of computational efficiency and memory usage? UniTSFace utilizes the ResNet architecture as its backbone and optimizes the parameters using the algorithmic average of the cosine-margin Softmax loss and the proposed USS loss. When compared to CosFace, the extra computational cost brought by USS loss is relatively small, as the computational consumption and memory usage mostly depend on the convolutional operations inherent to the selected network architecture and the resolutions of the input images. In experiments, we indeed found that the computational differences between these methods during both the testing and training stages are negligible.

**Conclusion.**

We propose the USS loss for deep face recognition by explicitly defining a unified threshold that separates all positive sample-to-sample pairs from negative ones. Though this unified threshold learned in the training stage cannot be directly applied to the testing stage, the model trained by USS is desired to extract more discriminative features and subsequently improve the face recognition performance in various testing scenarios, which have been demonstrated with extensive experiments.

Furthermore, the proposed USS loss can be effortlessly extended to the Marginal USS loss and can also be seamlessly combined with other sample-to-class losses. In our experiments, we have combined the USS loss with ArcFace and CosFace methods, and the combined approaches consistently surpass their respective individual counterparts. We denote the fusion of the CosFace and USS losses as "UniTSFace" which we have compared with other sophisticated combinations such as VPL, UNPG, and AnchorFace. The experimental results on multiple benchmark datasets further suggest the superiority of our UniTSFace.

In conclusion, we believe our USS loss provides the research community with a much more versatile and effective solution for face recognition tasks.

## Acknowledgement

This work was supported by the National Natural Science Foundation of China under Grants 82261138629 and 62006156, Guangdong Basic and Applied Basic Research Foundation under Grants 2023A1515010688 and 2022A1515012125, and Shenzhen Municipal Science and Technology Innovation Council under Grant JCYJ20220531101412030.

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

# Appendix for USS

## A  Inequalities about $L_{\text{naive}}$

We here present the detailed derivations for the inequalities about the naive loss $L_{\text{naive}}$. For any sample $\boldsymbol{X}^{(i)}$ captured from subject $i$, with feature $\boldsymbol{x}^{(i)} = \mathcal{M}(\boldsymbol{X}^{(i)})$, the naive loss $L_{\text{naive}}$ comprises of one positive sample-to-sample similarity and $N-1$ negative similarities,

$$L_{\text{naive}}(\boldsymbol{X}^{(i)}) = -\gamma g(\boldsymbol{x}^{(i)}, \boldsymbol{x}_*^{(i)}) + \frac{1}{N-1} \sum_{\substack{j=1 \\ j \neq i}}^{N} \gamma g(\boldsymbol{x}^{(i)}, \boldsymbol{x}_*^{(j)}), \tag{25}$$

where $\boldsymbol{x}_*^{(i)} = \mathcal{M}(\boldsymbol{X}_*^{(i)})$, $\boldsymbol{x}_*^{(j)} = \mathcal{M}(\boldsymbol{X}_*^{(j)})$, and $\boldsymbol{X}_*^{(i)}$ and $\boldsymbol{X}_*^{(j)}$ are randomly taken from the subject $i, j$, with $j \neq i$.

Using the inequality of arithmetic and geometric means, i.e., $\sqrt[n]{\prod_{i=1}^{n} a_i} \leq \frac{1}{n} \sum_{i=1}^{n} a_i$ for $a_i \geq 0$, we derive four inequalities about $L_{\text{naive}}$.

$$L_{\text{naive}}(\boldsymbol{X}^{(i)})$$

$$= -\gamma g(\boldsymbol{x}^{(i)}, \boldsymbol{x}_*^{(i)}) + \frac{1}{N-1} \sum_{\substack{j=1 \\ j \neq i}}^{N} \gamma g(\boldsymbol{x}^{(i)}, \boldsymbol{x}_*^{(j)})$$

$$= -\frac{N}{N-1} \left( \frac{N-1}{N} \gamma g(\boldsymbol{x}^{(i)}, \boldsymbol{x}_*^{(i)}) - \frac{1}{N} \sum_{\substack{j=1 \\ j \neq i}}^{N} \gamma g(\boldsymbol{x}^{(i)}, \boldsymbol{x}_*^{(j)}) \right) \tag{26}$$

$$= -\frac{N}{N-1} \left( \gamma g(\boldsymbol{x}^{(i)}, \boldsymbol{x}_*^{(i)}) - \frac{1}{N} \sum_{j=1}^{N} \gamma g(\boldsymbol{x}^{(i)}, \boldsymbol{x}_*^{(j)}) \right) \tag{27}$$

$$= -\frac{N}{N-1} \left[ \log \exp \left( \gamma g(\boldsymbol{x}^{(i)}, \boldsymbol{x}_*^{(i)}) \right) - \log \exp \left( \frac{1}{N} \sum_{j=1}^{N} \gamma g(\boldsymbol{x}^{(i)}, \boldsymbol{x}_*^{(j)}) \right) \right] \tag{28}$$

$$\leq -\frac{N}{N-1} \left[ \log \exp \left( \gamma g(\boldsymbol{x}^{(i)}, \boldsymbol{x}_*^{(i)}) \right) - \log \left( \frac{1}{N} \sum_{j=1}^{N} \exp \left( \gamma g(\boldsymbol{x}^{(i)}, \boldsymbol{x}_*^{(j)}) \right) \right) \right] \tag{29}$$

$$= -\frac{N}{N-1} \left( \log \frac{e^{\gamma g(\boldsymbol{x}^{(i)}, \boldsymbol{x}_*^{(i)})}}{\sum_{j=1}^{N} e^{\gamma g(\boldsymbol{x}^{(i)}, \boldsymbol{x}_*^{(j)})}} + \log N \right) \tag{30}$$

$$= -\frac{N}{N-1} \log \frac{e^{\gamma g(\boldsymbol{x}^{(i)}, \boldsymbol{x}_*^{(i)})}}{\sum_{j=1}^{N} e^{\gamma g(\boldsymbol{x}^{(i)}, \boldsymbol{x}_*^{(j)})}} - \frac{N \log N}{N-1}, \tag{31}$$

$$L_{\text{naive}}(\boldsymbol{X}^{(i)})$$

$$= -\gamma g(\boldsymbol{x}^{(i)}, \boldsymbol{x}_*^{(i)}) + \frac{1}{N-1} \sum_{\substack{j=1 \\ j \neq i}}^{N} \gamma g(\boldsymbol{x}^{(i)}, \boldsymbol{x}_*^{(j)})$$

$$= \gamma g(\boldsymbol{x}^{(i)}, \boldsymbol{x}_*^{(i)}) + \frac{1}{N-1} \sum_{\substack{j=1 \\ j \neq i}}^{N} \left( \gamma g(\boldsymbol{x}^{(i)}, \boldsymbol{x}_*^{(j)}) \right) - 2\gamma g(\boldsymbol{x}^{(i)}, \boldsymbol{x}_*^{(i)}) \tag{32}$$

$$= 2 \left[ \frac{1}{2} \left( \gamma g(\boldsymbol{x}^{(i)}, \boldsymbol{x}_*^{(i)}) + \frac{1}{N-1} \sum_{\substack{j=1 \\ j \neq i}}^{N} \gamma g(\boldsymbol{x}^{(i)}, \boldsymbol{x}_*^{(j)}) \right) - \gamma g(\boldsymbol{x}^{(i)}, \boldsymbol{x}_*^{(i)}) \right] \tag{33}$$

$$= 2 \left\{ \log \exp \left[ \frac{1}{2} \left( \gamma g(\boldsymbol{x}^{(i)}, \boldsymbol{x}_*^{(i)}) + \frac{1}{N-1} \sum_{\substack{j=1 \\ j \neq i}}^{N} \gamma g(\boldsymbol{x}^{(i)}, \boldsymbol{x}_*^{(j)}) \right) \right] - \log \exp \left( \gamma g(\boldsymbol{x}^{(i)}, \boldsymbol{x}_*^{(i)}) \right) \right\}$$

$$\tag{34}$$

$$\leq 2\left\{\log\left[\frac{1}{2}\left(\exp\left(\gamma g(\boldsymbol{x}^{(i)}, \boldsymbol{x}_*^{(i)})\right) + \exp\sum_{\substack{j=1 \\ j\neq i}}^{N}\frac{\left(\gamma g(\boldsymbol{x}^{(i)}, \boldsymbol{x}_*^{(j)})\right)}{N-1}\right)\right] - \log\exp\left(\gamma g(\boldsymbol{x}^{(i)}, \boldsymbol{x}_*^{(i)})\right)\right\} \tag{35}$$

$$= 2\left[\log\left(\exp\left(\gamma g(\boldsymbol{x}^{(i)}, \boldsymbol{x}_*^{(i)})\right) + \exp\sum_{\substack{j=1 \\ j\neq i}}^{N}\frac{\left(\gamma g(\boldsymbol{x}^{(i)}, \boldsymbol{x}_*^{(j)})\right)}{N-1}\right) - \log\exp\left(\gamma g(\boldsymbol{x}^{(i)}, \boldsymbol{x}_*^{(i)})\right) - \log 2\right] \tag{36}$$

$$= 2\log\left(1 + \frac{\exp\left(\sum_{\substack{j=1 \\ j\neq i}}^{N}\frac{\gamma g(\boldsymbol{x}^{(i)}, \boldsymbol{x}_*^{(j)})}{N-1}\right)}{\exp\left(\gamma g(\boldsymbol{x}^{(i)}, \boldsymbol{x}_*^{(i)})\right)}\right) - 2\log 2, \tag{37}$$

$$L_{\text{naive}}(\boldsymbol{X}^{(i)})$$

$$= -\gamma g(\boldsymbol{x}^{(i)}, \boldsymbol{x}_*^{(i)}) + \frac{1}{N-1}\sum_{\substack{j=1 \\ j\neq i}}^{N}\gamma g(\boldsymbol{x}^{(i)}, \boldsymbol{x}_*^{(j)})$$

$$= \frac{1}{N-1}\left(\sum_{\substack{j=1 \\ j\neq i}}^{N}\left(\gamma g(\boldsymbol{x}^{(i)}, \boldsymbol{x}_*^{(j)}) + \gamma g(\boldsymbol{x}^{(i)}, \boldsymbol{x}_*^{(i)})\right) - 2(N-1)\gamma g(\boldsymbol{x}^{(i)}, \boldsymbol{x}_*^{(i)})\right) \tag{38}$$

$$= \frac{2}{N-1}\left(\sum_{\substack{j=1 \\ j\neq i}}^{N}\frac{\gamma g(\boldsymbol{x}^{(i)}, \boldsymbol{x}_*^{(j)}) + \gamma g(\boldsymbol{x}^{(i)}, \boldsymbol{x}_*^{(i)})}{2} - (N-1)\gamma g(\boldsymbol{x}^{(i)}, \boldsymbol{x}_*^{(i)})\right) \tag{39}$$

$$= \frac{2}{N-1}\left(\sum_{\substack{j=1 \\ j\neq i}}^{N}\log\exp\frac{\gamma g(\boldsymbol{x}^{(i)}, \boldsymbol{x}_*^{(j)}) + \gamma g(\boldsymbol{x}^{(i)}, \boldsymbol{x}_*^{(i)})}{2} - (N-1)\log\exp\left(\gamma g(\boldsymbol{x}^{(i)}, \boldsymbol{x}_*^{(i)})\right)\right) \tag{40}$$

$$\leq \frac{2}{N-1}\left(\sum_{\substack{j=1 \\ j\neq i}}^{N}\log\frac{\mathrm{e}^{\gamma g(\boldsymbol{x}^{(i)}, \boldsymbol{x}_*^{(j)})} + \mathrm{e}^{\gamma g(\boldsymbol{x}^{(i)}, \boldsymbol{x}_*^{(i)})}}{2} - (N-1)\log\exp\left(\gamma g(\boldsymbol{x}^{(i)}, \boldsymbol{x}_*^{(i)})\right)\right) \tag{41}$$

$$= \frac{2}{N-1}\sum_{\substack{j=1 \\ j\neq i}}^{N}\left(\log\frac{\mathrm{e}^{\gamma g(\boldsymbol{x}^{(i)}, \boldsymbol{x}_*^{(j)})} + \mathrm{e}^{\gamma g(\boldsymbol{x}^{(i)}, \boldsymbol{x}_*^{(i)})}}{2} - \log\mathrm{e}^{\gamma g(\boldsymbol{x}^{(i)}, \boldsymbol{x}_*^{(i)})}\right) \tag{42}$$

$$= \frac{2}{N-1}\sum_{\substack{j=1 \\ j\neq i}}^{N}\log\frac{\mathrm{e}^{\gamma g(\boldsymbol{x}^{(i)}, \boldsymbol{x}_*^{(j)})} + \mathrm{e}^{\gamma g(\boldsymbol{x}^{(i)}, \boldsymbol{x}_*^{(i)})}}{2\mathrm{e}^{\gamma g(\boldsymbol{x}^{(i)}, \boldsymbol{x}_*^{(i)})}} \tag{43}$$

$$= \frac{2}{N-1}\sum_{\substack{j=1 \\ j\neq i}}^{N}\log\left[\frac{1}{2}\left(1 + \frac{\mathrm{e}^{\gamma g(\boldsymbol{x}^{(i)}, \boldsymbol{x}_*^{(j)})}}{\mathrm{e}^{\gamma g(\boldsymbol{x}^{(i)}, \boldsymbol{x}_*^{(i)})}}\right)\right] \tag{44}$$

$$= \frac{2}{N-1}\sum_{\substack{j=1 \\ j\neq i}}^{N}\log\left(1 + \frac{\mathrm{e}^{\gamma g(\boldsymbol{x}^{(i)}, \boldsymbol{x}_*^{(j)})}}{\mathrm{e}^{\gamma g(\boldsymbol{x}^{(i)}, \boldsymbol{x}_*^{(i)})}}\right) - 2\log 2, \tag{45}$$

and

$$\sum_{i=1}^{N}L_{\text{naive}}(\boldsymbol{X}^{(i)})$$

$$= \frac{1}{N-1}\left(\sum_{i=1}^{N}\sum_{\substack{j=1 \\ j\neq i}}^{N}\gamma g(\boldsymbol{x}^{(i)}, \boldsymbol{x}_*^{(j)}) - (N-1)\sum_{i=1}^{N}\gamma g(\boldsymbol{x}^{(i)}, \boldsymbol{x}_*^{(i)})\right) \tag{46}$$

$$= \frac{1}{N-1} \Big( \sum_{i=1}^{N} \sum_{\substack{j=1 \\ j \neq i}}^{N} \gamma g(\boldsymbol{x}^{(i)}, \boldsymbol{x}_*^{(j)}) - \sum_{i=1}^{N} \sum_{\substack{j=1 \\ j \neq i}}^{N} \gamma g(\boldsymbol{x}^{(j)}, \boldsymbol{x}_*^{(j)}) \Big) \tag{47}$$

$$= \frac{1}{N-1} \sum_{i=1}^{N} \sum_{\substack{j=1 \\ j \neq i}}^{N} \Big( \gamma g(\boldsymbol{x}^{(i)}, \boldsymbol{x}_*^{(j)}) - \gamma g(\boldsymbol{x}^{(j)}, \boldsymbol{x}_*^{(j)}) \Big) \tag{48}$$

$$= \frac{1}{N-1} \sum_{i=1}^{N} \sum_{\substack{j=1 \\ j \neq i}}^{N} \Big( \gamma g(\boldsymbol{x}^{(i)}, \boldsymbol{x}_*^{(j)}) + \gamma g(\boldsymbol{x}^{(j)}, \boldsymbol{x}_*^{(j)}) - 2\gamma g(\boldsymbol{x}^{(j)}, \boldsymbol{x}_*^{(j)}) \Big) \tag{49}$$

$$= \frac{2}{N-1} \sum_{i=1}^{N} \sum_{\substack{j=1 \\ j \neq i}}^{N} \Big( \log \exp \frac{\gamma g(\boldsymbol{x}^{(i)}, \boldsymbol{x}_*^{(j)}) + \gamma g(\boldsymbol{x}^{(j)}, \boldsymbol{x}_*^{(j)})}{2} - \log \exp \big( \gamma g(\boldsymbol{x}^{(j)}, \boldsymbol{x}_*^{(j)}) \big) \Big) \tag{50}$$

$$\leq \frac{2}{N-1} \sum_{i=1}^{N} \sum_{\substack{j=1 \\ j \neq i}}^{N} \Big( \log \frac{\mathrm{e}^{\gamma g(\boldsymbol{x}^{(i)}, \boldsymbol{x}_*^{(j)})} + \mathrm{e}^{\gamma g(\boldsymbol{x}^{(j)}, \boldsymbol{x}_*^{(j)})}}{2} - \log \exp \big( \gamma g(\boldsymbol{x}^{(j)}, \boldsymbol{x}_*^{(j)}) \big) \Big) \tag{51}$$

$$= \frac{2}{N-1} \sum_{i=1}^{N} \sum_{\substack{j=1 \\ j \neq i}}^{N} \Big[ \log \big( \mathrm{e}^{\gamma g(\boldsymbol{x}^{(i)}, \boldsymbol{x}_*^{(j)})} + \mathrm{e}^{\gamma g(\boldsymbol{x}^{(j)}, \boldsymbol{x}_*^{(j)})} \big) - \log \exp \big( \gamma g(\boldsymbol{x}^{(j)}, \boldsymbol{x}_*^{(j)}) \big) \Big] - 2N \log 2 \tag{52}$$

$$= \frac{2}{N-1} \sum_{i=1}^{N} \sum_{\substack{j=1 \\ j \neq i}}^{N} \log \Big( 1 + \frac{\mathrm{e}^{\gamma g(\boldsymbol{x}^{(i)}, \boldsymbol{x}_*^{(j)})}}{\mathrm{e}^{\gamma g(\boldsymbol{x}^{(j)}, \boldsymbol{x}_*^{(j)})}} \Big) - 2N \log 2. \tag{53}$$

## B   Marginal Sample-to-Sample Based Losses

We have derived three sample-to-sample based losses in the manuscript, i.e., USS loss, sample-to-sample based softmax, and BCE losses. We hereby present their respective marginal versions:

$$L_{\text{uss-m}}(\boldsymbol{X}^{(i)}) = \log \Big( 1 + \mathrm{e}^{-\gamma(g(\boldsymbol{x}^{(i)}, \boldsymbol{x}_*^{(i)})-m)+b} \Big) + \sum_{\substack{j=1 \\ j \neq i}}^{N} \log \Big( 1 + \mathrm{e}^{\gamma g(\boldsymbol{x}^{(i)}, \boldsymbol{x}_*^{(j)})-b} \Big), \tag{54}$$

$$L_{\text{soft-m}}(\boldsymbol{X}^{(i)}) = = -\log \frac{\mathrm{e}^{\gamma(g(\boldsymbol{x}^{(i)}, \boldsymbol{x}_*^{(i)})-m)}}{\sum_{j=1}^{N} \mathrm{e}^{\gamma g(\boldsymbol{x}^{(i)}, \boldsymbol{x}_*^{(j)})}}, \tag{55}$$

$$L_{\text{bce-m}}(\boldsymbol{X}^{(i)}) = \log \Big( 1 + \mathrm{e}^{-\gamma(g(\boldsymbol{x}^{(i)}, \boldsymbol{x}_*^{(i)})-m)+b_i} \Big) + \sum_{\substack{j=1 \\ j \neq i}}^{N} \log \Big( 1 + \mathrm{e}^{\gamma g(\boldsymbol{x}^{(i)}, \boldsymbol{x}_*^{(j)})-b_j} \Big). \tag{56}$$

The experimental evaluations of such marginal losses have been included in Sec. 4.3 of the manuscript.

## C   Training Details

In our work, we choose the cosine function to represent the similarity of two features, i.e.,

$$g(\boldsymbol{x}, \boldsymbol{x}_*) = \cos(\boldsymbol{x}, \boldsymbol{x}_*) = \frac{\langle \boldsymbol{x}, \boldsymbol{x}_* \rangle}{\|\boldsymbol{x}\| \|\boldsymbol{x}_*\|}, \quad \text{for } \forall \, \boldsymbol{x}, \boldsymbol{x}_* \in \mathcal{F}. \tag{57}$$

Following (30; 7), we use customized ResNets (such as ResNet-50, ResNet-100, and ResNet-200) as our backbone networks. We implement all models using Pytorch and train them using the SGD optimizer with a weight decay of 5e-4 and momentum of 0.9. We use $\gamma = 64$ for $L_{\text{uss}}$, $L_{\text{soft}}$, and $L_{\text{bce}}$ in all experiments. Note that, we use a combination of CosFace($m = 0.4$) and our USS($m = 0.1$) as UniTSFace in Tables 4 and 5.

For the face models (using ResNet-50 as the backbone) on CASIA-WebFace, we train them over 28 epochs with a batch size of 512. The learning rate starts at 0.1 and is reduced by a factor of 10 at the $16^{th}$ and $24^{th}$ epoch. All models in ablation and parameter study were trained on CASIA-WebFace.

In Table 1, the margin $m$ of $L_{\text{soft-m}}$, $L_{\text{bce-m}}$ and $L_{\text{uss-m}}$ are set to be 0.1. In Table 3, the margin $m$ of ArcFace and CosFace are set to be 0.5 and 0.4 respectively. The UniTSFace under the 'Small' protocol of MegaFace Challenge 1 in Table 4 and the models re-implemented in MFR Ongoing (the first ten rows in Table 5) were also trained on CASIA-WebFace.

For Glint360K, we train the models(ResNet-100) for 20 epochs using a batch size of 1024. Initially, the learning rate was set at 0.1, and a polynomial decay strategy (power=2) was applied to the learning rate schedule. The UniTSFace under the 'Large' protocol of MegaFace Challenge 1 (as shown in Table 4) was trained on Glint360K.

For WebFace4M, we train the models(ResNet-50) for 20 epochs using a batch size of 1024. The learning rate was initially set at 0.1, while a polynomial decay strategy (power=2) was applied to the learning rate schedule. The UniTSFace at the $12^{th}$ row of Table 5 was trained on WebFace4M.

In the case of WebFace42M, we train the models(ResNet-200) for 20 epochs, using a larger batch size of 4096. The learning rate linearly warmed up from 0 to 0.4 during the first epoch, followed by a polynomial decay (power=2) for the remaining 19 epochs. The UniTSFace at the last row of Table 5 was trained on WebFace42M.

## D   Megaface Challenge 1

We notice that the official MegaFace challenge website has been decommissioned and MegaFace data are no longer being distributed. However, despite these changes, we can still utilize the previously released data and development kit to evaluate the performance of trained models. Specifically, we have adopted the MegaFace testsuite provided by InsightFace (`https://github.com/deepinsight/insightface/tree/master/recognition/_evaluation_/megaface`), which also includes the official devkit.

