# OpenReview forum: "UniTSFace: Unified Threshold Integrated Sample-to-Sample Loss for Face Recognition"
_NeurIPS.cc/2023/Conference — NeurIPS 2023 poster_

### Official Review · Reviewer_m9uU · 2023-07-04

**Soundness:** 2 fair
**Presentation:** 2 fair
**Contribution:** 2 fair
**Rating:** 5
**Confidence:** 2

**Summary:**

I thank the authors for interesting paper and the effort in expanding the field of face recognition.
The paper proposes a new loss function USS (and its variants) for face recognition. The motivation is to jointly learn a threshold that can be used during the verification process. The authors show the efficacy of UniTSFace when USS is used jointly with ArcFace or CosFace (margin based softmax loss) in various evaluation sets.

**Strengths:**

- The performance gain in various datasets using UniTSFace is clearly shown.
- The derivation of USS loss starting from contrastive objective is interesting.
- The motivation of trying to learn one threshold that can be used in all evaluation dataset is interesting.

**Weaknesses:**

- The authors do not show whether learning the threshold t is really necessary. The readers cannot distinguish whether the performance gain comes from making the threshold t, a learnable parameter or it comes from the addition of the supervised contrastive learning objective which is a loss function that boosts the performance when combined with margin based softmax losses [1].
- The authors do not show whether the learned threshold is indeed better than the more common method which is to calculate the optimal threshold specific to the validation dataset.
- Figure 1 seems to suggest that despite the L_uss loss, the variance in threshold remains therefore, it is better to calibrate different threshold for different datasets.

[1] CoReFace: Sample-Guided Contrastive Regularization for Deep Face Recognition

**Questions:**

- It was not stated clearly in the text what unified refers to. If it is correct that the authors use the term unified because the learned threshold is used for all evaluation settings *(in contrast to the normal scenario where the threshold is computed for each dataset), then it would be good to say it in the beginning.
- Why does the unified threshold satisfy the equation 3? There may not exist t that satisfies this for a given dataset.
- How does UniTSFace compare with ArcFace + Supervised Contrasitve Loss (InfoNCE) and ArcFace + L_naive (which does not learn t). In such case, t is computed for each validation dataset as is customary for face recognition.


**Limitations:**

Yes. The limitation is discussed.

---

> ### Author Rebuttal · Authors · 2023-08-10
>
> **Q1. The authors do not show whether learning the $t$ is really necessary.**
>
> Thanks for the suggestion.
>
> We have compared the performance of $L_{\text{uss}}$ and $L_{\text{naive}}$ in **Table A**, it is evident that $L_{\text{uss}}$, by adding the $t$, achieves a remarkable enhancement over $L_{\text{naive}}$. We note that we dedicated extensive efforts to train the model with $L_{\text{naive}}$, but the model struggled to converge, resulting in inferior results.
>
> In **Tables A and B**, we further compared the performance of both $L_{\text{uss}}$ and $L_{\text{naive}}$, when employed in conjunction with the CosFace and ArcFace. Similarly, we observed that the combination of ArcFace/CosFace with $L_{\text{naive}}$ resulted in notably inferior outcomes compared to the baseline ArcFace/CosFace. This starkly contrasts with the more favorable performance achieved by pairing ArcFace/CosFace with $L_{\text{uss}}$. These results show that the $t$ plays a pivotal role in boosting performance.
>
> Moreover, we note that in Table 1 of our paper, we have compared our $L_{\text{uss}}$ with the other two baseline sample-to-sample losses, $L_{\text{soft}}$ and $L_{\text{bce}}$ (neither has a unified $t$). Our $L_{\text{uss}}$ again outperforms these two losses, which also shows the effectiveness of incorporating the $t$.
>
> **Q2. Whether the learned $t$ is indeed better than the more common method which is to calculate the optimal threshold specific to the validation dataset.**
>
> Thanks for the question and we will clarify this in the revision.
>
> Firstly, the learned $t$ is not directly used in testing. In our experiments, the optimal threshold is indeed calculated specific to the validation dataset as well as the testing criteria.
>
> Secondly, though the unified $t$ learned by the model on the training dataset cannot be directly applied to the validation dataset, the model itself has learned compact and discriminative features. Specifically, without this explicit $t$, the losses, $L_{\text{naive}}$, $L_{\text{soft}}$, and $L_{\text{bce}}$, only promote proximity among intra-subject features and discrepancy among inter-subject features for each subject. With this explicit $t$, our USS loss aims to learn features to distinguish all the positive sample-to-sample similarities from all the negative ones on the whole training dataset. In other words, the learning objective of USS loss is more stringent, the features learned using our USS loss are therefore expected to be more discriminative. The results reported in Table 1 of the paper, as well as in the **Tables A and B**, demonstrate the benefits of the features learned by adding the unified $t$.
>
> **Q3. Figure 1, the variance in threshold remains therefore, it is better to calibrate different threshold for different datasets.**
>
> As per our response to **Q2**, the optimal testing threshold is indeed calibrated for specific dataset.
>
> On the other hand, though there are variances in Fig. 1, we note that the distribution from $L_{\text{uss}}$ is much more compact than the other two counterparts, which qualitatively support our claim that the model has managed to learn more compact and discriminative features, due to the application of $L_{\text{uss}}$ loss to learn a unified threshold.
>
> **Q4. It was not stated clearly in the text what unified refers to.**
>
> Thanks for the advice and we will clarify this in the revision.
>
> In the case of training, the unified threshold $t$ distinguishes our $L_{\text{uss}}$ from the other sample-to-sample loss such as $L_{\text{naive}}$, $L_{\text{soft}}$, and $L_{\text{bce}}$, which only care about the subject-level separation: i.e., intra-subject similarity is higher than inter-subject similarity. They do not pursue that, for all subjects, all the positive sample-to-sample similarity should be larger than all the negative sample-to-sample similarity. The $t$ in our $L_{\text{uss}}$, however, explicitly requires that all the positive sample-to-sample similarity should be larger than all the negative sample-to-sample similarity for all samples. We, therefore, name our loss as a unified threshold integrated loss.
>
> In the case of testing, the learned $t$ on the training set cannot be directly applied to the testing datasets. The optimal testing threshold will be calculated based on different datasets as well as the application scenarios.
>
> **Q5. Why does the unified $t$ satisfy the equation 3? There may not exist a $t$ that satisfies this for a given dataset.**
>
> We agree that there might not exist a unified $t$ that satisfies Eq. (3) for a given dataset, which have noise samples, or wrong labels.
>
> Ideally, if the dataset is clean and the face model is well-trained, all positive sample-to-sample similarity should be close to 1 and negative similarity should be close to -1. Then the $t$ will exist.
>
> Naturally, our USS loss is built on the assumption that there exists such a unified $t$ for a given dataset, then it is our objective to find the unified threshold satisfying Eq. (3) in the training of a facial model. We have demonstrated that the unified $t$ can be learned after the training of the model by adopting the USS loss, in lines 141-148, and the unified $t$ is evolved into the learnable parameter bias $b=\gamma t$ in the USS loss.
>
> Even though the unified $t$ may not exist, the learning process towards this ideal condition still enables the model to learn more compact and discriminative features than the other losses, which can lead to better performance on various testing datasets.
>
> **Q6. Compare UniTSFace with ArcFace + InfoNCE / + L_naive.**
>
> As per the reviewer's request, we have conducted the related experiments and reported the results in **Table A**. It is clear that ArcFace + $L_{\text{naive}}$ produced significantly inferior results than the plain ArcFace, while ArcFace + InfoNCE improves the performance of ArcFace, it is still outperformed by our UniTSFace, demonstrating the effectiveness of our method.

---

> > ### Comment · Reviewer_m9uU · 2023-08-16
> >
> > I appreciate the comprehensive reply from the authors. I have carefully reviewed the response. I no longer have any questions, and my issues have been resolved. I encourage the authors to further refine the paper based on reviewer feedback, and I'd like to once again thank them for their diligent efforts.

---

> > > ### Author Response · Authors · 2023-08-17
> > >
> > > Thanks for raising the scores and the valuable suggestions; we will further refine the paper according to all reviewers' comments, when submitting the final version.

---

### Official Review · Reviewer_XuAx · 2023-07-05

**Soundness:** 3 good
**Presentation:** 3 good
**Contribution:** 2 fair
**Rating:** 4
**Confidence:** 5

**Summary:**

The paper proposes a unified threshold integrated sample-to-sample loss for face recognition, which addresses the limitations of existing methods in exploring the cross-sample relationship and setting a unified threshold. The proposed loss function achieves exceptional performance on multiple benchmark datasets.

**Strengths:**

1) The paper introduces an approach called UniTSFace, which combines a unified threshold with a sample-to-sample loss for face recognition. This approach addresses the limitations of existing methods and proposes a new loss function that achieves exceptional performance on multiple benchmark datasets.
2) The experimental results of the paper demonstrate significant improvement, especially compared to the baseline, on highly discriminative datasets such as MFR.

**Weaknesses:**

1) It is unnecessary to use complex symbolic formulas to express simple concepts, such as in lines 105 to 124.
2) Adding a sample-to-sample loss in the field of face recognition is not novel enough.
3) The training method of using a fixed threshold may appear to be consistent with testing, but it actually conflicts with it. In testing, different false acceptance rates (FAR) are used on different scenarios, meaning that for the same model, different thresholds will be used to determine the predicted results.

**Questions:**

Please refer to the Weaknesses section.

---

> ### Author Rebuttal · Authors · 2023-08-10
>
> **Q1. It is unnecessary to use complex symbolic formulas to express simple concepts, such as in lines 105 to 124.**
>
> Thanks for the comments. The reviewer appears to be well-versed in face recognition research and reckons that these formulas present a simple concept. However, it is essential for us to cater to a wider range of readers, including beginners in this field, and accurately and clearly present preliminary concepts in a comprehensive manner.
>
> Moreover, through these formulas, we aim to mathematically present our motivation for this work, i.e., explicitly learning a unified threshold to distinguish the positive sample-to-sample similarities from the negative ones on the training set, which is absent in existing sample-to-sample losses.
>
> Additionally, these notations are consistently used in sections 3.1, 3.2, and 3.3 to formulate different losses, namely $L_{\text{naive}}$, $L_{\text{soft}}$, $L_{\text{bce}}$, and $L_{\text{uss}}$. Removing these notations/formulas directly would adversely affect the presentation of these losses and lead to misunderstandings.
>
> Therefore, we believe it is crucial to retain the formulas in lines 105-124 to maintain a coherent and informative paper that caters to both experienced researchers and newcomers to the field of face recognition.
>
> **Q2. Adding a sample-to-sample loss in the field of face recognition is not novel enough.**
>
> Sample-to-sample based loss has been a crucial research topic in deep face recognition for the past decades. Researchers have dedicated substantial efforts to this area and have introduced prestigious losses and methods such as DeepID2, Triplet Loss, and (N+1)-Tuplet Loss, just to name a few. The majority of them aim to maximize inter-subject discrepancy while minimizing intra-subject distances, which may need a meticulous sampling/paring step for every mini-batch. Moreover, in face verification tasks, a single threshold is required to distinguish positive facial pairs from negative ones. Unfortunately, none of the aforementioned methods incorporate such an explicit constraint.
>
> To address this limitation, we propose the USS, which explicitly incorporates a learnable unified $t$ during the training process. By encouraging to separate positive and negative image pairs by the $t$, we expect that the model is able to learn more discriminative features, which thereafter can perform better on the unseen testing sets.
>
> Additionally, through our derivations, the $t$ can be embedded in the bias term $b=\gamma t$. After the model is trained, we are able to directly investigate the value of this $t$, which can help us to quantitively and qualitatively understand what the model is learning, as shown in Fig. 1 in the paper.
>
> Furthermore, the USS can be effortlessly extended to the Marginal USS loss and can be jointly used with other sample-to-class losses. In our experiments, we have combined the USS loss with CosFace as UniTSFace, which surpassed other sophisticated combinations such as VPL, UNPG, and AnchorFace. In fact, as the reviewer pointed out, "**the experimental results of the paper demonstrate significant improvement**".
>
> Reviewer nR9r agrees with us: “**It is interesting to focus on the threshold for distinguishing positive from negative pairs, which is paid less attention.**” Reviewer qAqH also stated that “**The advantages of the USS loss over other losses are numerous**”. Overall, we assert that our USS is novel and provides the research community with a much more general and effective solution for face recognition tasks.
>
> **Q3. The training method of using a fixed $t$ may appear to be consistent with testing, but it actually conflicts with it. In testing, different FAR are used on different scenarios, meaning that for the same model, different thresholds will be used.**
>
> We thank the reviewer for pointing this out. We will clarify this part in the revised version.
>
> Firstly, we note that the threshold $t$ is not fixed during training but a learnable parameter embedded in $b=\gamma t$. For different training datasets, the learned $t$ might be different.
>
> Secondly, it is true that different thresholds will be selected based on various criteria such as different FAR. This however, is not conflicting with our goal of learning a unified threshold in our USS loss. We explain the reasons as follows:
>
> 1. The training and testing stages in the context of deep face recognition are two completely separate stages. During the training stage, a face model is supervised and trained using either sample-to-class or sample-to-sample loss functions. Once the model is trained, it will solely be employed to extract latent facial features for testing samples (which may not necessarily belong to the subjects in the training set). The extracted features will be used to evaluate the performance of the trained face model. In other words, the performance of a face model essentially relies on effective feature learning, which is, features corresponding to the same subject are brought closer together, while features from different subjects are pushed apart. In testing, based on the features extracted by the learned deep face models, different thresholds will be determined for best performance in different scenarios.
>
> 2. We propose to explicitly learn a unified threshold $t$ through our USS loss.  The training process towards the unified threshold ensures that the model not only promotes proximity among intra-subject features and discrepancy among inter-subject features, but also explicitly encourages all positive sample-to-sample similarities to exceed all negative sample-to-sample similarities. By imposing this more stringent constraint, the features learned using our USS loss are anticipated to be more discriminative.
>
> 3. Though the learned unified $t$ might not be directly used in various testing scenarios, the ultimate goal of separating the negative from positive pairs using a unified threshold is consistent in both training and testing.

---

> > ### Comment · Reviewer_XuAx · 2023-08-13
> >
> > Thanks for the authors' response. I agree with the explanations provided by the authors for Q1 and Q2, but I still haven't received a very clear answer to the most important Q3. We can still only see the necessity of widening the distance between intra-class and inter-class distances, but for the main point of this paper: a unified threshold, there still hasn't been convincing evidence presented to demonstrate its necessity.

---

> > > ### Author Response · Authors · 2023-08-14
> > >
> > > **Q3'. We can still only see the necessity of widening the distance between intra-class and inter-class distances, but for the main point of this paper: a unified threshold, there still hasn't been convincing evidence presented to demonstrate its necessity.**
> > >
> > >
> > > We thank the reviewer for agreeing our explanations provided for Q1 and Q2.
> > > As for the concerns raised by Q3, we apologize for not clearly stating the necessity of learning the unified threshold in our first rebuttal to Q3. we would like to clarify as follows:
> > >
> > > **1)  "We can still only see the necessity of widening the distance between intra-class and inter-class distances."**
> > >
> > > We propose the unified threshold integrated sample-to-sample (USS) loss to learn the unified threshold $t$ on the training dataset. By integrating such an objective into the loss functions, the model trained by our USS loss is guided to produce more discriminative features than the model trained by other losses.
> > >
> > > The reviewer agrees that the necessity of widening the distance between intra-class and inter-class distances, which is actually consistent with the learning of an unified threshold, i.e., we need a threshold to separate them and the combination with margins can further improve the performance. As our USS loss is expected to separate all positive from negative pairs by the explicit threshold $t$, which is a **stricter** constraint than simply widening the distance between intra-class and inter-class distances.
> > >
> > > **2)  "for the main point of this paper: a unified threshold, there still hasn't been convincing evidence presented to demonstrate its necessity."**
> > >
> > > We highlight that we have quantitatively and qualitatively demonstrated the necessity of learning the unified threshold and we provide the evidence below.
> > >
> > > Firstly, in Table A, significant improvements have been achieved by $L_{\text{uss}}$ over $L_{\text{naive}}$, which does not learn an explicit threshold. Similar improvements can also be observed between $ArcFace + L_{\text {naive}}$ and $ArcFace + L_{\text{uss}}$ in Table A, as well as $CosFace + L_{\text{naive}}$ and $CosFace + L_{\text{uss}}$ in Table B.
> > >
> > > These improvements clearly show the necessity of learning the unified threshold.
> > >
> > > Secondly, in Table 1 of the paper, we have compared $L_{\text{uss}}$ and $L_{\text{bce}}$, and their marginal extensions  $L_{\text{uss-m}}$ and $L_{\text{bce-m}}$, the improvement achieved by our USS over the sample-to-sample BCE are also significant.
> > >
> > > We notice that the only difference between $L_{\text{uss}}$ and $L_{\text{bce}}$ is the objective of a unified threshold $t$ and the results further show the necessity of learning the unified threshold.
> > >
> > > Thirdly, we depicted the optimal threshold distributions of sample-to-sample Softmax loss, BCE loss, and our USS loss in Figure 1 of the paper. We can clearly observe that the distribution learned by our USS has the smallest variance and the learned unified threshold $t = 0.4896$ lies around the median in the interquartile range.
> > >
> > > This qualitative illustration also suggests the necessity of learning the unified threshold.

---

### Official Review · Reviewer_qAqH · 2023-07-08

**Soundness:** 4 excellent
**Presentation:** 4 excellent
**Contribution:** 4 excellent
**Rating:** 8
**Confidence:** 5

**Summary:**

This paper proposes UniTSFace, a new approach to face recognition that uses a unified threshold integrated sample-to-sample loss (USS loss). The USS loss features a unified threshold for distinguishing positive from negative pairs and can be enhanced with an auxiliary margin. The authors show that the USS loss can be integrated with sample-to-class based losses and evaluate its effectiveness on various benchmark datasets, demonstrating its suitability for real-world applications. The contributions of this work include introducing the USS loss, demonstrating that it can learn a unified threshold, and showing that it can be enhanced with an auxiliary margin and is compatible with existing sample-to-class based losses.

In addition, UniTSFace outperforms state-of-the-art methods on multiple benchmark datasets, including the Megaface 1 dataset, and the proposed approach is evaluated on the large-scale WebFace42M dataset, which contains 42.5 million images of 2 million identities, demonstrating its efficacy for real-world applications.

**Strengths:**

1. The advantages of the USS loss over other loss functions are numerous. First, it is highly efficient and can work seamlessly with sample-to-class-based losses. Second, it overcomes the limitations of previous sample-to-sample losses by explicitly incorporating a learnable threshold that separates positive and negative pairs. Third, it achieves state-of-the-art performance on multiple benchmark datasets, demonstrating its effectiveness in real-world face recognition applications.
2. The experiments in this paper are solid and have been conducted on the largest-scale dataset WebFace42M, achieving the state-of-the-art performance on the most challenging benchmark dataset MFR in the academic track, and the comparisons are fair.
3. The proposed method is original
4. There are no significant flaws in the experiments conducted in this work.

**Weaknesses:**

The paper does not explicitly mention any weaknesses or limitations of the proposed UniTSFace approach or the USS loss function.

**Questions:**

1. The paper mentions that the unified threshold is limited to a sample-to-sample loss. Have you considered extending the unified threshold to sample-to-class loss, and if so, what are the potential benefits and challenges of doing so?
2. The paper mentions that UniTSFace achieves state-of-the-art performance on multiple benchmark datasets. However, how does UniTSFace compare to other state-of-the-art methods in terms of computational efficiency and memory usage?


**Limitations:**

The paper does not explicitly mention any limitations

---

> ### Author Rebuttal · Authors · 2023-08-10
>
> **Q1.The paper mentions that the unified threshold is limited to a sample-to-sample loss. Have you considered extending the unified threshold to sample-to-class loss, and if so, what are the potential benefits and challenges of doing so?**
>
> We thank the reviewer for pointing this out. We are actually in the process of integrating the unified threshold into a sample-to-class loss, which we term USC loss here in coordination with USS loss for convenience.
>
> The sample-to-class-based losses are conventionally built on Softmax loss in face recognition, including the well-known Large Margin Softmax loss, normalized Softmax loss, cosine-margin and angular-margin based Softmax loss, such losses only learn a class proxy conveyed in the weight vector, instead of exploring the cross-sample relationship. To incorporate the unified threshold into the sample-to-class Softmax loss, it is straightforward that we directly introduce a unified threshold:
> $$L_{\text{usc}}(\textbf X^{(i)})=
> \log(1+e^{-\gamma\ g(\textbf x^{(i)}, \textbf c^{(i)})+b})
> +\sum_{j\neq i\atop j=1}^N\log(1+e^{\gamma\ g(\textbf x^{(i)}, \textbf c^{(j)}) - b}),$$
> where $b = \gamma t$ is a constant to be learned, $t$ hereby is the desired unified threshold, and $\textbf c^{(i)}$ is the feature proxy of the $i$-th class.
>
> Since the unified threshold in USC provides a stricter constraint than the original normalized Softmax loss, i.e., we anticipate that such a unified threshold can separate all the positive sample-to-class similarities from the negative ones. The USC loss is expected to outperform the original normalized Softmax counterpart in face recognition.
>
> Additionally, the USC is essentially a sample-to-class-based classification loss, it can naturally be used in general object classification tasks.
>
> On the other hand, we must admit that the USC loss has its own shortcomings in face recognition. Different from the sample-to-sample loss (such as $L_{\text{naive}}$, $L_{\text{uss}}$) used in this submission, it cannot directly explore the relationships among the facial samples in the training of the facial model, as it only optimizes the class proxy and sample features during training. The key challenge towards robust face recognition is how to learn a representative class proxy $\textbf c^{(i)}$ that can benefit from the abundant training facial samples. Therefore, as we stated in the conclusion of the manuscript, effectively combing the two approaches into one framework will be our future work.
>
>
> **Q2. The paper mentions that UniTSFace achieves state-of-the-art performance on multiple benchmark datasets. However, how does UniTSFace compare to other state-of-the-art methods in terms of computational efficiency and memory usage?**
>
> We thank the reviewer for pointing this out. UniTSFace utilizes the ResNet architecture as its backbone and optimizes the parameters using the algorithmic average of the cosine-margin Softmax loss and the proposed USS loss.
>
> In the inference stage, the trained UniTSFace model is solely a convolutional ResNet used to extract latent facial features for testing images. Naturally, the computational consumption and memory usage totally depend on the convolutional operations inherent to the selected ResNet architecture and the resolutions of the input images used during testing.
>
> Since all the methods under comparison in Tables 4 and 5 adopt the same ResNet-50, ResNet-100, and ResNet-200 backbones and the same input images for testing, the differences in computational and memory usage are negligible. This is affirmed in **Table C**, where we present the inference runtime across various methods. The Table clearly indicates that the variations in inference times among different methods are negligible, measured in fractions of a millisecond.
>
> Contrastingly, in the training stage, different models employ different loss functions to train the convolutional ResNet backbone, resulting in a diverse computational overhead and memory usage. Theoretically,
>
> 1. when comparing to deep models based on the purely sample-to-class losses, such as CosFace and ArcFace, UniTSFace requires additional $O(N)$ logarithmic/multiply-addition operations attributable to the USS loss. The joint USS loss entails additional memory usage to accommodate $N$ facial features, where $N$ denotes the number of subjects within the training dataset.
> 2. when comparing to deep models based on the purely sample-to-sample losses, e.g., $L_{\text{soft}}$ or $L_{\text{bce}}$, UniTSFace only requires $O(N)$ extra logarithmic/multiply-addition operations to account for the cosine-marginal Softmax loss.
> 3. when comparing to deep models trained with a combination of the two kinds of losses like VPL, UNPG, and AnchorFace, the computational requirements and memory consumption of our UniTSFace remain at par.
>
> In experiments, since the forward and backward propagation of convolutional operations (including the activations functions and normalization layers) accounts for the majority of computations and memory usage, we found that the computational difference between these methods during the training stage is minimal as well. As reported in **Table C**, the memory usage (in GB) and training speeds (in seconds per batch, i.e., seconds per 512 images) across different methods similarly exhibit minimal discrepancies.

---

### Official Review · Reviewer_nR9r · 2023-07-12

**Soundness:** 3 good
**Presentation:** 3 good
**Contribution:** 2 fair
**Rating:** 6
**Confidence:** 4

**Summary:**

This paper presents an interesting idea by focusing on the unified threshold of distinguishing positive from negative pairs in face recognition. Although face recognition has undergo a big step towards real application driven by deep learning, this submission has some new proposal for sample-to-sample based loss with unified threshold. Experiments on typical facial image datasets show the effectiveness.


===============
After reading the authors' rebuttals to my concerns and dicussions, I upgrade my rating as "weak accept".

**Strengths:**

+It is interesting to focus on the threshold for distinguishing positive from negative pairs, which is paid less attention.

+The derivation of the USS loss makes sense by defining the  upperbound of the naive loss.

+Experiments by deploying USS into ArcFace and CosFace show the effectiveness of the proposed USS. The threshold range is narrowed, which shows the certainty of threshold is improved.


**Weaknesses:**

-From Table 1, it shows that BCE loss is much worse than USS loss. This seems strange and I concern if the experiment is wrong for this loss. The reasons should be discussed.

-From Fig.1, the threshold range is narrowed compared to other losses, but I concern if it is correct to claim the word "unified".

-Face recognition models are explosive increased with deep learning. A number of advanced methods are developed. The necessity of this submission can be discussed in the paper. The contribution of the proposed method to the community can be discussed by combining with previous advanced face recognition models.

**Questions:**

In inference stage, how to determine the threshold? pre-computed? It is better to clarify this point because the title is unified theshold, which gives me the first expression about the pre-trained threshold.

**Limitations:**

Not observed.

---

> ### Author Rebuttal · Authors · 2023-08-10
>
> **Q1. From Table 1, it shows that BCE loss is much worse than USS loss. This seems strange and I concern if the experiment is wrong for this loss. The reasons should be discussed.**
>
> Firstly, we believe that the results are correct and we want to emphasize that the results are not cherry-picked. For a fair comparison, in Table 1, we adopted the same experimental setting when we trained the model using three different losses: $L_{\text{soft}}$, $L_{\text{bce}}$, and $L_{\text{uss}}$.
>
> Specifically, during our experiments, we found that the model is easy to converge using $L_{\text{soft}}$ and $L_{\text{uss}}$, and the training process is stable, which leads to more favorable results at the end.
>
> However, we found that the model is extremely hard to converge when using the $L_{\text{bce}}$. Despite many efforts such as prolonged iterations, have been dedicated, the convergence is still problematic. We reckon that $L_{\text{bce}}$ has respective one bias term for each subject, resulting in $N$ different explicit thresholds as $b_i = \gamma t_i$, which lead to instable training. In contrast, the $L_{\text{uss}}$ has only one explicit threshold and $L_{\text{soft}}$ does not integrate any explicit threshold.
>
> **Q2. From Fig.1, the threshold range is narrowed compared to other losses, but I concern if it is correct to claim the word "unified".**
>
> We thank the reviewer for pointing this out.
>
> Firstly, our theoretical objective is to learn a unified threshold that satisfying Eq. (3) during the training of a facial model, which is consistent with the requirement of the testing. Therefore, we first assume the existence of such a unified threshold, and then propose the unified threshold integrated sample-to-sample (USS) loss.
>
> We have proven through our analysis in lines 141-148 of the paper that, ideally, a model trained by USS could learn a unified threshold for the training dataset.
>
> However, we must admit that achieving this ideal goal is subjective to the model capacity, training hyper-parameters, and even the training dataset itself, which are all independent from our USS loss.  For example, if the backbone network only uses one single linear neural layer, our USS loss definitely cannot guarantee a unified threshold either. In Fig. 1, however, using the same backbone architecture and training hyper-parameters, our USS loss are able to achieve a more compact threshold distribution than the other losses, which suggests the superiority of imposing the unified threshold and is consistent with our expectation.
>
> Therefore, we believe it is reasonable to retain the word "unified".
>
>
> **Q3. Face recognition models are explosive increased with deep learning. A number of advanced methods are developed. The necessity of this submission can be discussed in the paper. The contribution of the proposed method to the community can be discussed by combining with previous advanced face recognition models.**
>
> We thank the advice from the reviewer.
>
> Significant advancements in deep face recognition have been introduced to the community in the past decade, including sample-to-class-based diagrams, sample-to-sample-based diagrams, and the combinations of the two diagrams. However, none of them can explicitly learn a unified threshold to separate the positive sample-to-sample similarities from the negative similarities among all samples in the whole training dataset. An optimal threshold separating all positive sample-to-sample pairs from the negative ones is also in demand during testing.
>
> In our submission, we propose the USS loss to explicitly learn a unified threshold for the training dataset. Though this unified threshold learned in the training stage cannot be directly applied to the testing stage, the model trained by USS is desired to extract more discriminative features and subsequently improves the face verification performance in various testing scenarios, which have been demonstrated through our extensive experiments.
>
> Furthermore, the proposed USS loss can be effortlessly extended to the Marginal USS loss and can also be seamlessly jointly used with other sample-to-class losses. In our experiments, we have combined the USS loss with ArcFace and CosFace methods, and the combined approaches consistently surpass their respective individual counterparts. We denote the fusion of the CosFace and USS losses as "UniTSFace," which we have compared with other sophisticated combinations such as VPL, UNPG, and AnchorFace.  The experimental results on multiple benchmark datasets further suggest the superiority of our UniTSFace.
>
> In conclusion, we believe our USS loss provides the research community with a much more versatile and effective solution for face recognition tasks.
>
> **Q4. In inference stage, how to determine the threshold? pre-computed? It is better to clarify this point because the title is unified threshold, which gives me the first expression about the pre-trained threshold.**
>
> We thank the reviewer for this suggestion. We will clarify this in the revised version.
>
> Yes, the threshold learned by the training stage cannot be directly used in testing. In the testing/inference stage, we first extract the features using trained models for all testing images, and then the threshold is determined according to the specific testing criteria. For example, when reporting the 1:1 verification accuracy on LFW, CFP-FP, AgeDB in Table 5, 10-fold validation is used. We first select the threshold which achieves the highest accuracy in the first 9 folds, and then adopt this threshold to calculate the accuracy in the leave-out fold.

---

> > ### Comment · Reviewer_nR9r · 2023-08-12
> > **Concern on the inference stage**
> >
> > Thanks for the authors' detailed response. About Q4, the authors said that in testing phase,  the features should be extracted for all testing images and 10 fold validation is used. But this seems not that practical in testing, and in real application, we may not collect all testing images. How to predict the face ID per image?

---

> > > ### Author Response · Authors · 2023-08-12
> > >
> > > We thank the reviewer for the question and we would like to clarify as follows.
> > >
> > >
> > > 1) **Face recognition is different from general image classification.**
> > >
> > > The general image recognition tasks are closed-set recognition, such as digital zip code recognition in MNIST and natural image classification in ImageNet. Both the training and testing categories are fixed in such tasks, which means that each testing image must be classified into one of the existing classes. For example, every test image in MNIST must and will be classified into a number between 0 and 9 according to the classification probabilities.
> > >
> > > However, real-world face recognition is often an open-set recognition task, i.e., the training images and testing images are not always limited to the same classes (identities/individuals). For example, the used training dataset CASIA-WebFace has only about 10,575 identities while globalized multi-racial (GMR), one of the testing sets, contains 242,143 identities (>> the 10575 training IDs). Therefore, it is evident that we cannot directly input a testing image into the trained model and predict its ID according to the classification probabilities.
> > >
> > > 2) **Face recognition can be divided into face verification and face identification.**
> > >
> > > Face recognition tasks can be divided into 1:1 verification and 1:n identification.
> > >
> > > In 1:1 verification, we will be given two face images (one probe image and one template image) and we are required to predict whether these two images belong to the same ID. 1:1 verification is the same as iPhone Face ID and Google Pixel Face Unlock, we have a template face image saved in the phone and we need to distinguish whether the saved template face and the probe face in front of the camera are from the same individual, or not.
> > >
> > > In 1:n identification, we will be given a probe face with an unknown ID and a gallery set that stores a set of faces of known individuals, we are required to pick up the right ID for the probe face image from the given gallery set. 1:n identification is often been seen in movies where the police take a picture of the person of interest and compare it with a local database.
> > >
> > > 3) **How to predict the face ID per image?**
> > >
> > > In 1:1 verification, we use the trained face model to extract features for both the probe image with an unknown ID and the template image with a known ID. Subsequently, we compare the feature similarity with a pre-defined unified threshold $\hat{t}$, if the similarity is greater than the $\hat{t}$, the two images are assumed to have the same ID, vise versa.
> > >
> > > In 1:n identification, we use the trained face model to extract features for the probe face image (unknown ID) as well as all the images in the gallery set (known IDs). The probe face will be assigned to the ID of the gallery image that has the highest feature similarity with the probe image.
> > >
> > > 4) **Q4 seems not that practical in testing, In a real application, we may not collect all testing images.**
> > > 1. For LFW, CFP-FP, and AgeDB in Table 5, we report the 1:1 verification accuracy with 10-fold validation. For IJB-C, we report True Accept Rate (TAR) at False Accept Rate (FAR)= 1e-4 and 1e-5. For the MFR benchmarks, we report TARs at FAR=1e-4 for the Mask and Children test sets, and TARs at FAR=1e-6 for the GMR test sets. For the MegaFace Challenge 1, we report Rank1 accuracy for identification and TAR at FAR=1e-6 for verification.
> > > These metrics have been widely and standardly used to evaluate the performance of different facial models, we hereby follow the same settings for a fair comparison.
> > >
> > > 2. We agree that we cannot collect all testing images in real applications. Therefore, many efforts have been dedicated to collecting larger datasets to mimic real-world scenarios, for example, LFW (2007) only has 13,233 images from 5749 identities, while GMR (2021) contains 1,624,305 images from 242,143 identities.
> > >
> > > 3. We notice that for most real applications, we only need to extract and compare features for the probe image and the template image in 1:1 verification, or to extract and compare features for the probe image and the faces in gallery sets in 1:n identification, which are practical.
> > >
> > >
> > > Hope our clarification answers the question raised. And of course, we are open to continue discussions if the reviewer has further questions.

---

> > > > ### Comment · Reviewer_nR9r · 2023-08-13
> > > > **Concern on the inference stage**
> > > >
> > > > In the authors' first rebuttal, you claimed "In the testing/inference stage, we first extract the features using trained models for all testing images, and then the threshold is determined according to the specific testing criteria."
> > > > I am concerned about this sentence. In my opinion, the threshold should be determined by a validation set. In testing phase, the test should be sample by sample, and you should not use all testing data (because in real application, for each query image, its correct ID is predicted and matched). If you use all testing data, it first violates the real-application protocol and second you have seen the test data because the threshold is also a parameter.

---

> > > > > ### Author Response · Authors · 2023-08-13
> > > > >
> > > > > We apologize for the concerns raised by our first rebuttal. We would like to address them as follows:
> > > > >
> > > > > **1)	 “In my opinion, the threshold should be determined by a validation set. In testing phase, the test should be sample by sample, and you should not use all testing data”.**
> > > > >
> > > > > We agree with the reviewer. This description is exactly what we did when reporting the 1:1 verification performance. We explain the whole testing protocol using the well known LFW as an example.
> > > > >
> > > > > LFW, widely used in face verification, contains 13,233 images from 5749 identities. The official testing protocol 1) first evenly pairs the images to 3000 positive and 3000 negative pairs, resulting in 6000 pairs in total. 2) randomly divide the 6000 pairs into 10-fold of 600 (300 positive and 300 negative) pairs. 3) determine the optimal threshold $\hat{t}_i$ on the rest 9-folds according to the ROC curve. 4) using the determined threshold $\hat{t}_i$ to evaluate on the left-out fold and report the 1:1 verification accuracy for this left-out fold. 5) repeat this process and compute all 10 optimal thresholds $\hat{t}_i $ for all 10-fold cross-validation and report the averaged 10-fold verification accuracy.
> > > > >
> > > > > The whole testing protocol of LFW clearly includes a "validation" procedure to determine the threshold $\hat{t}_i $ and a "testing" procedure that compares the $\hat{t}_i $ with the feature similarity of each pair. In other words, **we did not use all testing data to select the threshold $\hat{t}_i$.**
> > > > >
> > > > > **2)	The sentence, “we first extract the features using trained models for all testing images, and then the threshold is determined according to the specific testing criteria”, in the first rebuttal, brings ambiguity.**
> > > > >
> > > > > We apologize for the ambiguity brought by this sentence.
> > > > >
> > > > > Sticking to the LFW as the example, we intend to use "all testing images" to denote the 6000 pairs of images. It is logical and convenient to extract all the image features at the very beginning of the testing protocol.
> > > > >
> > > > > And then "the threshold is determined according to the specific testing criteria", for which we highlight that we did not use all 6000 pairs of features at once, but selected the respective threshold $\hat{t}_i $ with cross-validation for each left-out set.
> > > > >
> > > > > Therefore, although we have extracted the features for all testing images, we did not use them to compute a global optimal threshold to report the performance, but following the cross-validation protocol to first determine the threshold $\hat{t}_i$ for each fold and report the accuracy, which definitely not violates the real-application protocol either.
> > > > >
> > > > > Moreover, although we said "we first extract the features using trained models for all testing images, and then the threshold is determined according to the specific testing criteria", we actually uploaded the trained model to the challenge website and the results are computed by their end.
> > > > >
> > > > >
> > > > > **3)	What happens with the other datasets that use TAR at FAR=1e-K as the criteria?**
> > > > >
> > > > > We report the 1:1 verification accuracy with 10-fold validation for not only LFW but also CFP-FP, and AgeDB, therefore, the testing of these datasets also precisely matches "the threshold should be determined by a validation set. In the testing phase, the test should be sample by sample, and you should not use all testing data".
> > > > >
> > > > > For other datasets such as IJB-C and MFR that use TARs at FAR=1e-4, 1e-5, or 1e-6 as the criteria, neither of them contains an explicit validation set. But their validation process can be regarded as determining the threshold according to the FAR, while their testing process is to use the determined threshold from FAR to report the TAR.
> > > > >
> > > > > At last, though there is ambiguity in our previous description, we notice that for all datasets that reports the 1:1 verification accuracy, we did not break the principle of pair-wise comparison, we exactly followed the standard testing protocol.
> > > > >
> > > > > Moreover, all results on the Tables 1,2,3 and 5 in the paper, as well as the Table A and B in the first rebuttal, are computed by the MFR-challenge website, following the standard testing protocol as well.

---

### Author Rebuttal · Authors · 2023-08-10

We thank all reviewers for appreciating the state-of-the-art performance of our UniTSFace; especially the recommendations from reviewers qAqH and nR9r that “It is interesting to focus on the threshold for distinguishing positive from negative pairs, which is paid less attention,” and “the advantages of our USS loss over other losses are numerous”. We provide point-to-point responses to the main concerns raised by the reviewers. We here present the experimental results requested by the reviewers and attach them in the PDF file.


**Table A. Comparisons of different losses/methods on the MFR-Ongoing dataset. We note that model trained with Lnaive encountered challenges in convergence and exhibited the least favorable results.**
|     Method                 |     MR-ALL    |     IJB-C    |     LFW      |     CFP      |     Age      |
|----------------------------|---------------|--------------|--------------|--------------|--------------|
|     $L_{\text{naive}}$                 |     0.0       |     0.35     |     50.0     |     50.0     |     50.0     |
|     $L_{\text{uss}}$                   |     38.43     |     72.20    |     99.40    |     96.51    |     94.05    |
|     ArcFace                |     42.21     |     48.49    |     99.31    |     97.07    |     94.51    |
|     ArcFace   + InfoNCE    |     45.47     |     88.00    |     99.40    |     97.11    |     94.71    |
|     ArcFace   + $L_{\text{naive}}$     |     0.48      |     3.52     |     98.21    |     80.42    |     84.13    |
|     ArcFace   + $L_{\text{uss}}$       |     48.76     |     89.06    |     99.58    |     97.40    |     94.73    |
|     ArcFace   + $L_{\text{soft-m}}$    |     46.03     |     88.16    |     99.45    |     97.20    |     95.03    |
|     ArcFace   + $L_{\text{bce-m}}$     |     17.28     |     65.10    |     96.38    |     74.40    |     82.81    |
|     ArcFace   + $L_{\text{uss-m}}$     |     48.92     |     89.56    |     99.40    |     97.22    |     95.20    |

**Table B. Comparisons of different losses in combination with CosFace on the MFR-Ongoing dataset.**
|     Method                 |     MR-ALL    |     IJB-C    |     LFW      |     CFP      |     Age      |
|----------------------------|---------------|--------------|--------------|--------------|--------------|
|     CosFace                |     45.12     |     56.65    |     99.36    |     97.30    |     94.98    |
|     CosFace   + $L_{\text{naive}}$     |     1.82      |     13.03    |     98.98    |     95.47    |     93.05    |
|     CosFace   + $L_{\text{uss}}$       |     49.75     |     89.62    |     99.41    |     96.78    |     95.30    |
|     CosFace   + $L_{\text{soft-m}}$   |     47.90     |     88.71    |     99.46    |     97.12    |     95.50    |
|     CosFace   + $L_{\text{bce-m}}$     |     15.53     |     61.53    |     96.15    |     73.20    |     80.36    |
|     CosFace   + $L_{\text{uss-m}}$      |     50.28     |     89.84    |     99.41    |     97.35    |     95.13    |

**Table C. Comparisons of different methods/losses in terms of the Inference Time (in millisecond), Training Memeory usage (in GB), and Training Speeds (in seconds per batch i.e., seconds per 512 images). All experiments are conducted on the same machine with Intel(R) Xeon(R) Silver 4110 CPU @ 2.10GHz; TITAN RTX, 24GB.**
|     Method        |     Inference Time     (ms)    |     Training Memory   (G)    |     Training Speeds   (s/batch, s/512images)    |
|-------------------|--------------------------------|------------------------------|-------------------------------------------------|
|     ArcFace       |     14.42                      |     9.20                     |     0.93                                        |
|     CosFace       |     14.25                      |     9.20                     |     0.92                                        |
|     $L_{\text{soft}}$         |     14.50                      |     9.19                     |     0.96                                        |
|     $L_{\text{bce}}$          |     14.42                      |     9.19                     |     0.97                                        |
|     $L_{\text{uss}}$          |     14.33                      |     9.22                     |     0.97                                        |
|     VPL           |     14.25                      |     9.19                     |     0.97                                        |
|     UNPG          |     14.66                      |     9.24                     |     0.96                                        |
|     AnchorFace    |     14.25                      |     9.24                     |     0.99                                        |
|     UniTSFace     |     14.60                      |     9.22                     |     0.97                                        |

---

### Decision · Program_Chairs · 2023-09-21

**Decision:**

Accept (poster)

**Comment:**

Most of the reviewers gave positive ratings on this submission while the Reviewer XuAx had some concerns on writing and fixed threshold. After carefully read the submission, comments and rebuttals, the AC agree with most of the reviewers and suggest accepting this submission.